# Inactivation of *Arid1a* in the endometrium is associated with endometrioid tumorigenesis through transcriptional reprogramming

Yohan Suryo Rahmanto [1 ✉], Wenjing Shen[1], Xu Shi [2], Xi Chen[2], Yu Yu [1,7], Zheng-Cheng Yu[1], Tsutomu Miyamoto[1], Meng-Horng Lee[3], Vivek Singh [1], Ryoichi Asaka[1], Geoffrey Shimberg[1], Michele I. Vitolo[4], Stuart S. Martin[4], Denis Wirtz [3], Ronny Drapkin [5], Jianhua Xuan[2], Tian-Li Wang[1,6 ✉] & Ie-Ming Shih[1,6 ✉]

Somatic inactivating mutations of *ARID1A*, a SWI/SNF chromatin remodeling gene, are prevalent in human endometrium-related malignancies. To elucidate the mechanisms underlying how *ARID1A* deleterious mutation contributes to tumorigenesis, we establish genetically engineered murine models with *Arid1a* and/or *Pten* conditional deletion in the endometrium. Transcriptomic analyses on endometrial cancers and precursors derived from these mouse models show a close resemblance to human uterine endometrioid carcinomas. We identify transcriptional networks that are controlled by Arid1a and have an impact on endometrial tumor development. To verify findings from the murine models, we analyze *ARID1A*$^{WT}$ and *ARID1A*$^{KO}$ human endometrial epithelial cells. Using a system biology approach and functional studies, we demonstrate that *ARID1A*-deficiency lead to loss of TGF-β tumor suppressive function and that inactivation of ARID1A/TGF-β axis promotes migration and invasion of *PTEN*-deleted endometrial tumor cells. These findings provide molecular insights into how *ARID1A* inactivation accelerates endometrial tumor progression and dissemination, the major causes of cancer mortality.

[1] Sidney Kimmel Comprehensive Cancer Center and Department of Pathology, Johns Hopkins Medical Institutions, Baltimore, MD 21205, USA. [2] Bradley Department of Electrical and Computer Engineering, Virginia Polytechnic Institute and State University, Arlington, VA 22203, USA. [3] Department of Chemical and Biomolecular Engineering, Physical Sciences-Oncology Center, and Institute for NanoBioTechology, Johns Hopkins University, Baltimore, MD 21218, USA. [4] Marlene and Stewart Greenebaum National Cancer Institute Cancer Center and Department of Physiology, University of Maryland, Baltimore, MD 21201, USA. [5] Department of Obstetrics and Gynecology, University of Pennsylvania, Philadelphia, PA 19104, USA. [6] Department of Gynecology and Obstetrics, Johns Hopkins Medical Institutions, Baltimore, MD 21287, USA. [7] Present address: School of Pharmacy and Biomedical Sciences, Curtin Health Innovation Research Institute, Curtin University, Perth, WA 6102, Australia. ✉email: suryoysr@gmail.com; tlw@jhmi.edu; ishih@jhmi.edu

A berrant chromatin remodeling resulting from somatic mutations in genes encoding chromatin remodeling complexes has emerged as a major epigenetic modification contributing to cancer development. Efforts in cancer genome sequencing have uncovered frequent somatic mutations in genes encoding subunits of ATP-dependent chromatin remodeling complexes, most notably the SWI/SNF nucleosome remodeling complex[1–4]. Somatic mutations in the *ARID1A* subunit of the SWI/SNF complex frequently occur in human cancers. In neoplasms arising from endometrium, including endometrioid or clear cell carcinoma of the uterus and ovary, up to 50% of tumors harbor mutations in *ARID1A*[5–11]. *ARID1A* mutations are mostly frameshift or nonsense, resulting in loss of protein expression and function, a pattern characteristic of tumor suppressor genes[12]. Given the important functional roles of ARID1A, better understanding of how *ARID1A* inactivation contributes to tumor development is critical to improve treatment in *ARID1A* mutated cancers.

The incidence of endometrial carcinoma of the uterus is rising worldwide, and is expected to become a significant contributor to cancer-related morbidity and mortality in women. Currently, endometrial carcinoma affects 62,000 women annually and claims the lives of more than 12,000 women each year in the United States[13]. Although most endometrial cancers are diagnosed at early, treatable stages, late diagnosis of endometrial cancers at advanced stages remains challenging to treat. The molecular genetic landscape of endometrial carcinoma has been comprehensively elucidated by the Cancer Genome Atlas (TCGA), and high frequencies of somatic mutations have been detected in well-known cancer driver genes including *ARID1A* (33%), *PTEN* (65%), *PIK3CA* (53%), and *CTNNB1* (30%)[7,14]. Endometrial carcinoma is classified into different subtypes with endometrioid and serous carcinoma being the most common representing 80–85% and 5–10% of newly diagnosed cases, respectively. Previous studies reported a high co-occurrence of *ARID1A* and *PTEN* (or *PIK3CA*) mutations primarily in the endometrioid subtype of endometrial carcinoma[7,11]. Somatic *PTEN* mutation is detected in ~16% of endometrial atypical hyperplasia lesions, the precursors of endometrial cancers; however, most endometrial hyperplasia lesions are indolent and infrequently progress to endometrioid carcinoma, unless other molecular genetic events are acquired[12,15]. Complete loss of ARID1A expression or clonal *ARID1A* mutation is rare in atypical endometrial hyperplasia while up to one-third of endometrioid carcinomas lose ARID1A expression or harbor somatic *ARID1A* mutations. Moreover, loss of ARID1A expression in endometrial biopsy or curettage is associated with a significantly increased risk and FIGO stages of uterine endometrioid carcinoma in subsequent hysterectomies[16]. These above findings suggest that ARID1A participates in progression rather than initiation of endometrioid carcinoma of the uterus.

The phenotypes associated with *PTEN* deletion in uterine tissues has been elucidated by conditional deletion of *Pten* in mouse uterus using the progesterone receptor promoter-driven expression of Cre recombinase (Pgr-Cre approach). Non-invasive endometrial cancer was observed in this genetically engineered mouse model, supporting a key role(s) of *PTEN* in endometrial pathogenesis[17]. However, progesterone receptor is also expressed in stromal cells and myometrium[17], raising the question of whether the non-epithelial components contribute to the mouse tumor phenotypes.

Genetically engineered mouse models that closely recapitulate steps of genetic alterations in human endometrial tumor are valuable for studying pathogenesis and for testing new approaches for cancer prevention, early detection and treatment[18]. Here, we use a Pax8-Cre strategy which allows doxycycline-induced conditional knockout of *Arid1a* and/or *Pten* in epithelial component of endometrium. Pax8 is a lineage specific transcription factor in the müllerian epithelium of uterus and fallopian tube but not ovary. The murine models created in this study accurately recapitulate tumor initiation, progression, and invasion of uterine endometrioid carcinoma. Using this model, we elucidate the molecular mechanisms in the pathway crosstalk between Arid1a and TGF-β signaling. We then further verify the importance of TGF-β in mediating ARID1A function on isogenic human endometrial cell lines with or without *ARID1A* deletion. Together, our results shed lights on how loss of *ARID1A* promotes invasion and metastasis and may inform future translational research aimed at improving outcomes in women with endometrioid carcinoma of the uterus.

## Results

**Arid1a loss in endometrial epithelium enhances tumor progression**. To determine the phenotype of ARID1A inactivation in endometrioid tumors, a murine model with conditional deletion of *Arid1a* in the uterine epithelium was generated by crossing an *Arid1a*^*flox/flox*^ strain with a previously established Pax8-Cre mouse strain[19]. The Pax8-Cre mouse contains a reverse tetracycline-regulated transactivator under the control of the murine Pax8 promoter (Pax8-rtTA) and a tetracycline-dependent Cre recombinase (TetO-Cre), resulting in a murine model with expression of doxycycline-activated Cre recombinase specifically in uterine epithelial cells (Fig. 1a). In human uterine endometrioid carcinomas, ARID1A and PTEN are frequently mutated or deleted (Fig. 1b)[7,20,21]. ARID1A and PTEN mutations co-occur frequently in the same tumors (Fig. 1b). Accordingly, we established a conditional Pax8-Cre Pten knockout mouse line and an *Arid1a/Pten* double knockout line. As a result, a total of three murine models were established, designated *iAD* (*Arid1a*^*flox/flox*^/*Pten*^*WT*^), *iPD* (*Arid1a*^*WT*^/*Pten*^*flox/flox*^), and *iPAD* (*Arid1a*^*flox/flox*^/*Pten*^*flox/flox*^) (Fig. 1a). Gene knockout efficiency in the endometrium of these murine models was robust, as evidenced by undetectable Arid1a immunoreactivity in uterine epithelial cells in *iAD* and *iPAD* mice (Supplementary Fig. 1A, C). Specific inactivation of Pten in *iPD and iPAD* mice was also confirmed by immunohistochemistry (Supplementary Fig. 1B, C).

Simultaneous deletions of *Arid1a* and *Pten* in *iPAD* mice resulted in grossly visible uterine tumors 6 weeks after doxycycline-induced knockout, while *iAD* mice or *iPD* mice (with single gene knocked out) did not present any gross tumors at the same time (Fig. 1c). In contrast, enlarged uteri containing endometrial tumors were present in *iPAD* mice. Local and peritoneal disseminated tumor nodules reflecting advanced stages were observed in doxycycline-treated *iPAD* mice within 6 weeks (Fig. 1c).

We performed histological examination on *iAD* mice and did not observe any microscopic abnormalities in uteri or other organs during a prolonged observation period (24 weeks, n = 110; Fig. 2a, b). At 2 weeks after doxycycline-induced knockout, *iPD* mice displayed uterine endometrial intraepithelial neoplasia (EIN), a pre-cancerous stage of invasive endometrioid carcinoma, in 17 of 20 mice (85%, Fig. 2a, b). Pax8 immunoreactivity was observed in all epithelial cells of mouse uterus, regardless of benign or neoplastic (Fig. 2b, c). The progression of these premalignant lesions in *iPD* mice was relatively slow or stagnant, as similar EIN distributions was detected at week 2 and week 6 after induced *Pten* deletion (Fig. 2a). In contrast, a highly-penetrant, early onset endometrial carcinoma was observed in all *iPAD* mice (n = 17) exhibiting a confluent glandular growth pattern 2 weeks after doxycycline administration (Fig. 2a–c). These tumors progressed rapidly to highly invasive carcinoma

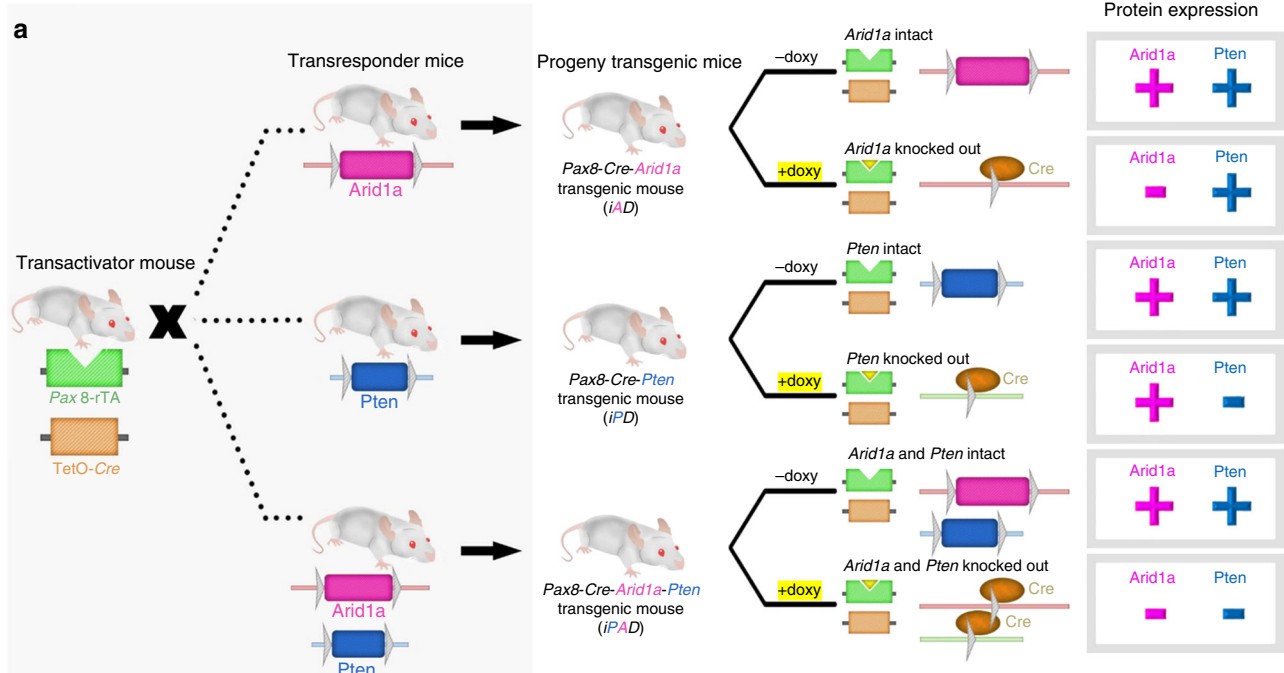

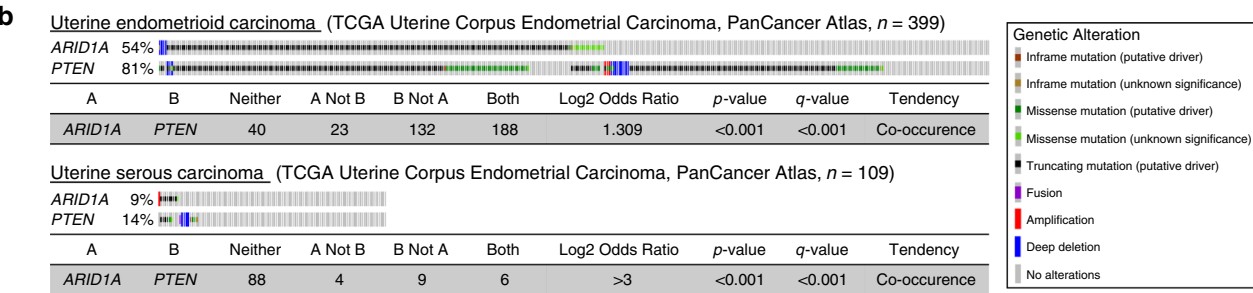

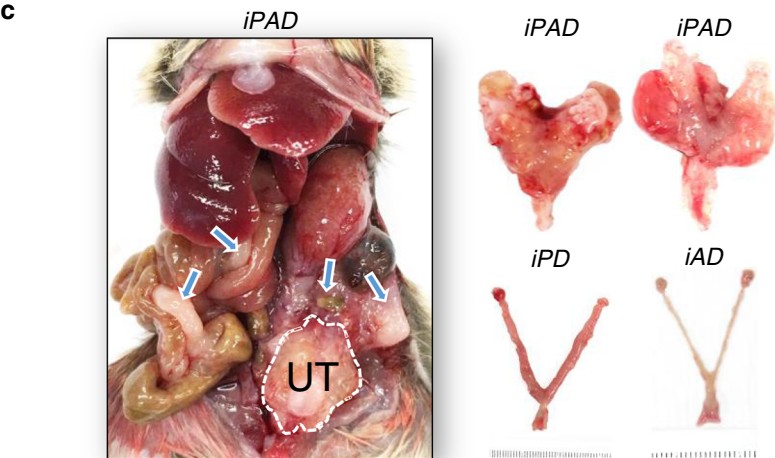

**Fig. 1 Arid1a deletion promotes endometrial cancer invasion in a Pax8-driven conditional knockout mouse model. a** The strategy in generating tissue-specific, conditional *Arid1a* knockout mouse models. Mice expressing Pax8-driven Cre (Pax8 rTA; TetO-Cre recombinant) were crossed with *Arid1a*^flox/flox (magenta color) or *Pten*^flox/flox mice (blue color). Flox-mediated gene recombination is activated by Tetracycline (yellow triangle) or its analog, doxycycline, which induces expression of Cre recombinase (orange color) in Pax8-expressing (green color) tissues. Cre subsequently mediates excision of the floxed gene segment of interest. rTA, reverse tetracycline-controlled transactivator; TetO, tetracycline operator; Cre, Cre recombinase; doxy, doxycycline. **b** Reported *ARID1A* and *PTEN* alterations in uterine endometrioid carcinoma and uterine serous carcinoma subtypes of the TCGA uterine endometrial carcinoma PanCancer Atlas dataset. *p*-values were determined by one-sided Fischer exact test. *q*-values were determined by Benjamini–Hochberg FDR correction. **c** Representative images of *iAD*, *iPD*, and *iPAD* uteri 6 weeks after doxycycline treatment. Local and peritoneal disseminated tumors (blue arrows) were only observed in *iPAD* mice. UT, uterine.

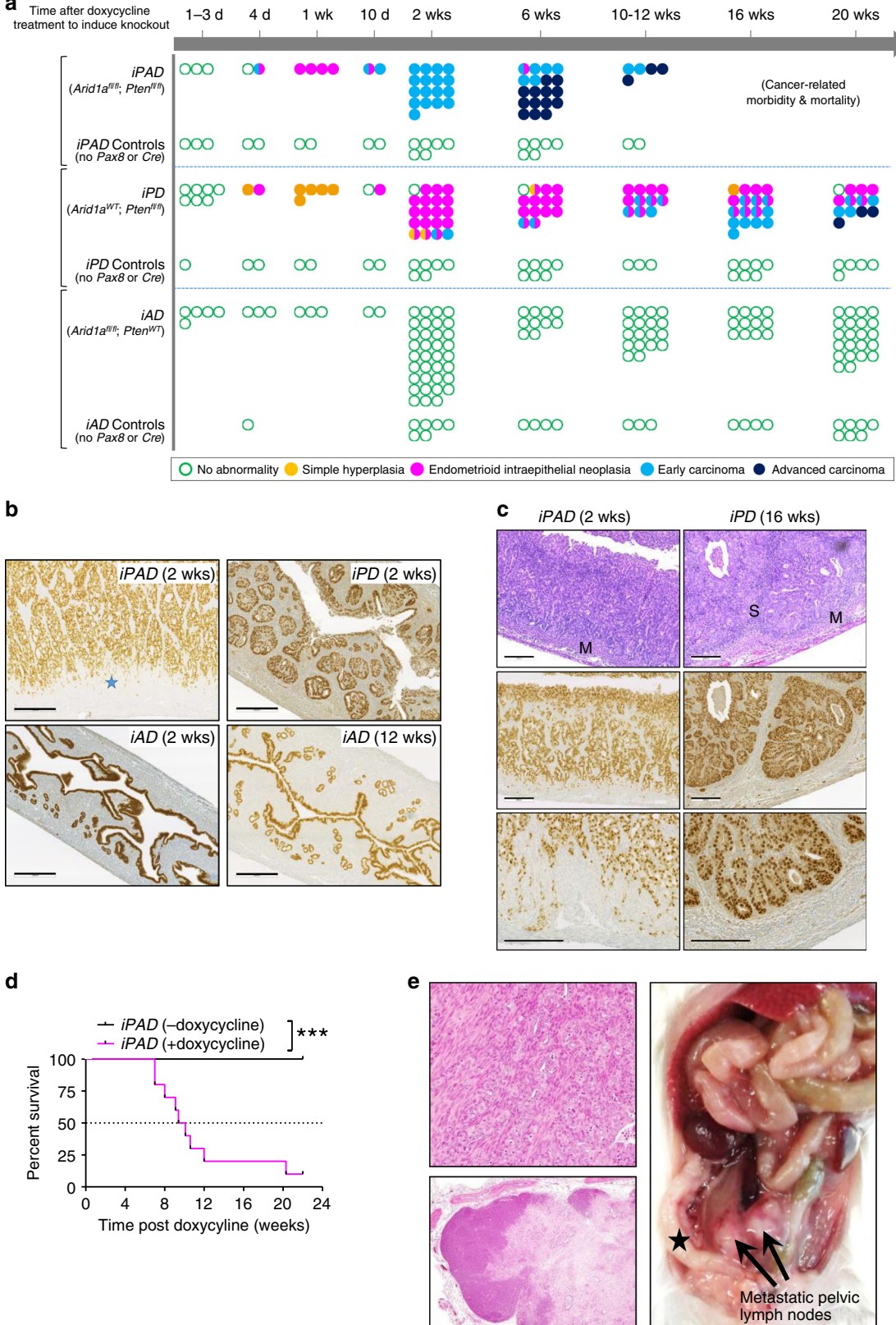

with morphological and clinical features similar to advanced endometrioid carcinoma. Myometrial and angiolymphatic invasions followed by tumor dissemination to the intraperitoneal cavity, and lymph node metastasis, which are characteristics of advanced endometrioid carcinoma, were observed in tumors from *iPAD* mice 6 weeks after doxycycline injection (Fig. 2a). By

comparison, myometrial invasion was found in only 3 of 30 (10%) *iPD* mice after more than 16 weeks of *Pten* deletion (Fig. 2a, c). All mice with the same transgene, but without doxycycline administration, were free of tumor at all time points examined, indicating that there was minimal leakage of the Pax8-Cre induction system (Fig. 2a). Doxycycline-treated *iPAD* mice

**Fig. 2 Histopathological characterization of endometrial cancer invasion in *Arid1a* conditional knockout mouse model. a** Diagram summarizing timeline and the corresponding phenotypes of mice with doxycycline-induced gene knockout in the endometrial epithelium. Pathological assessment was not performed on *iPAD* mice at weeks 16 and 20 due to cancer-related morbidity and mortality. All *iPAD* control mice were alive and active at both 16 and 20 weeks. Each circle represents an individual uterine sample. d, day; wk, week. *iPAD*: inducible *Pten* and *Arid1a* deletion; *iPD*: inducible *Pten* deletion; *iAD*: inducible *Arid1a* deletion. **b** Representative Pax8 immunohistochemistry (expressed in all epithelial cells of mouse uterine epithelium) highlighting the morphology of normal and lesional endometrial epithelial cells of *iAD*, *iPD*, and *iPAD* mice. Mice were treated with doxycycline for the period indicated at the top right corner. Blue star indicates tumor invasion into stroma. Wks, weeks. Scale bars, 300 µm. **c** Representative H&E (top 2 images) and Pax8 immunoreactivity (bottom 4 images) comparing morphological features between *iPAD* and *iPD* carcinomas at 2 weeks and 16 weeks. S, stromal; M, muscle. Scale bars, 200 µm. **d** Kaplan–Meier survival plot of *iPAD* mice with (magenta line) or without (black line) doxycycline-induced deletion. The dotted line indicates median survival. ***$p < 0.001$; log-rank test. **e** Metastasis of endometrial carcinoma to peritoneal lymph nodes. Left: gross photo from an *iPAD* mice 4 weeks after doxycycline treatment. Arrows indicate lymph nodes with tumor metastasis. Photomicrographs: H&E stained slides show invasive endometrial carcinoma in the right uterus (asterisk) (upper panel) and metastatic carcinoma within a representative lymph node (bottom panel).

had a significantly shortened median survival time of 9.75 weeks ($p < 0.001$, log-rank test), with the majority of mice succumbing to their disease by 20 weeks (Fig. 2d). Lymph nodes metastasis was observed in 30% of mice (Fig. 2e). In contrast, all doxycycline-treated *iPD* mice were alive and active at 20 weeks (Fig. 2a).

Individual examination of bilateral uteri of *iPAD* mouse at 2 weeks and 6 weeks post doxycycline-induced gene deletion revealed that non-invasive endometrial carcinoma has progressed to invasive carcinoma in all mice (Supplementary Fig. 2A). Necropsy of *iPAD* mice at 6 weeks post doxycycline treatment showed no macroscopic or microscopic abnormalities in other organs. Histopathological examination indicated that the kidneys from representative *iPAD* mice remained intact and tumor-free (Supplementary Fig. 2B), while endometrioid-like tumors were found throughout the uterus (Supplementary Fig. 2C).

Since the loss of Pten is associated with activation in the Pi3k and Akt pathway, we evaluated expression levels of p-Akt (Ser473) by immunohistochemistry in representative mouse uteri from all 3 mouse cohorts. Two weeks after the knockout, p-Akt immunoreactivity was detectable in EIN in *iPD* and *iPAD* but not in the endometrium of *iAD* mice (Supplementary Fig. 3). These data provide support that Pi3k activation occurs in mice with *Pten* deletion.

***iPAD* tumors molecularly recapitulate human endometrial cancer.** To identify the molecular features and differentially expressed genes in *iPAD* and *iPD* mouse tumors, we performed transcriptome profiling using RNA-seq. A list of 1156 genes showed a significant difference in mRNA expression levels between *iPAD* and *iPD* tumors, suggesting that these genes could be related to *Arid1a* gene deletion (FDR < 0.05, Benjamini–Hochberg FDR correction, and fold change >1.5) (Supplementary Data 1). Ingenuity pathway analysis (IPA) demonstrated IFNG, TNF, TGFB1, CTNNB1, and TP53 as the top upstream network regulators among the differentially expressed genes of the Arid1a-associated mouse transcriptome (Fig. 3a and Supplementary Data 2).

To assess molecular relevance of the mouse tumors to corresponding human tumors, we compared the mouse transcriptome data from *iPAD* and *iPD* uterine tumors with the transcriptome data from 7 major human carcinomas from The Cancer Genome Atlas (TCGA) PanCancer Atlas database, which were selected based on their histological similarity to endometrioid carcinomas, such as a confluent glandular growth pattern. In addition, uterine serous carcinoma was also included in analysis. By performing Principal Component and Euclidean Distance analyses, we found that *iPAD* and *iPD* tumors were most similar to human uterine endometrioid carcinoma, as reflected by the shortest Euclidean distance (Fig. 3b, c). Gene set enrichment analysis (GSEA)[22] was performed on TGFB1 target signature

molecules[23] against all of the *iPAD/iPD* mouse transcriptome shown in Supplementary Data 1. This analysis demonstrated that the TGF-β gene set was over-represented in the Arid1a-regulated transcriptome identified in mouse tumors (FDR = 0.000, Benjamini–Hochberg FDR correction) (Fig. 3d, top).

We next performed RNA-seq analysis on two isogenic pairs of human endometrial epithelial cell cultures with or without *ARID1A* gene, designated as wild-type ($ARID1A^{WT}$) and knockout ($ARID1A^{KO}$) cells. These pairs of $ARID1A^{WT}$ and $ARID1A^{KO}$ cell lines were established from human endometrial epithelium using a CRISPR/Cas9 genome editing[24]. Both $ARID1A^{WT}$ and $ARID1A^{KO}$ cells harbored wild-type *PTEN* and expressed a similar level of PTEN protein (Supplementary Fig. 4A). Our analysis identified 893 differentially expressed genes that were consistently regulated by ARID1A in both biological duplicates and regardless of whether it was grown under normal growth or serum-starved conditions (FDR < 0.05, Benjamini–Hochberg FDR correction, and fold change >1.5) (Supplementary Data 3). We identified 112 differentially expressed genes were common between the mouse and the human transcriptomes (Fig. 3e). Gene ontology analysis revealed that both the mouse and the human transcriptomes have common GO enrichment terms, such as oxidation-reduction, cell adhesion, cell proliferation, and cell migration processes in biological process GO category (Supplementary Data 5) and protein binding and oxidoreductase activity in molecular function GO category (Supplementary Data 5). IPA analysis of 893 ARID1A-associated genes from the human transcriptome showed that the upstream regulatory networks were similar to those identified in the murine tumor models (Fig. 3a and Supplementary Data 4). Among them, TGFB1 and TP53 networks were shared between ARID1A-regulated transcriptomes identified in murine tumors and human endometrial epithelial cells. GSEA analysis of ARID1A-associated human transcriptome against TGFB1 target signature molecules[23] revealed an enrichment of TGF-β signaling in $ARID1A^{WT}$ cells (FDR < 0.002, Benjamini–Hochberg FDR correction) (Fig. 3d, bottom). Our mouse endometrial tumors showed an inverse enrichment of TGFB1 target signature molecules as compared to the human endometrial epithelial cell cultures (Fig. 3d). This opposite observation was likely due to stromal TGF-β signaling in the mouse endometrial tumors, as the intra-tumoral stromal cells are *ARID1A* wild-type and responsive to TGF-β stimulation. Thus, the TGF-β signaling is enriched in *iPAD* tumor but not in the cell culture model.

**ARID1A loss abrogates TGF-β signaling and promotes invasiveness.** Our molecular studies discussed above illustrated TGF-β signaling as a component of an ARID1A-regulated network. Moreover, previous studies suggest TGF-β signaling can be regulated by ARID1A[25,26], which prompted us to interrogate

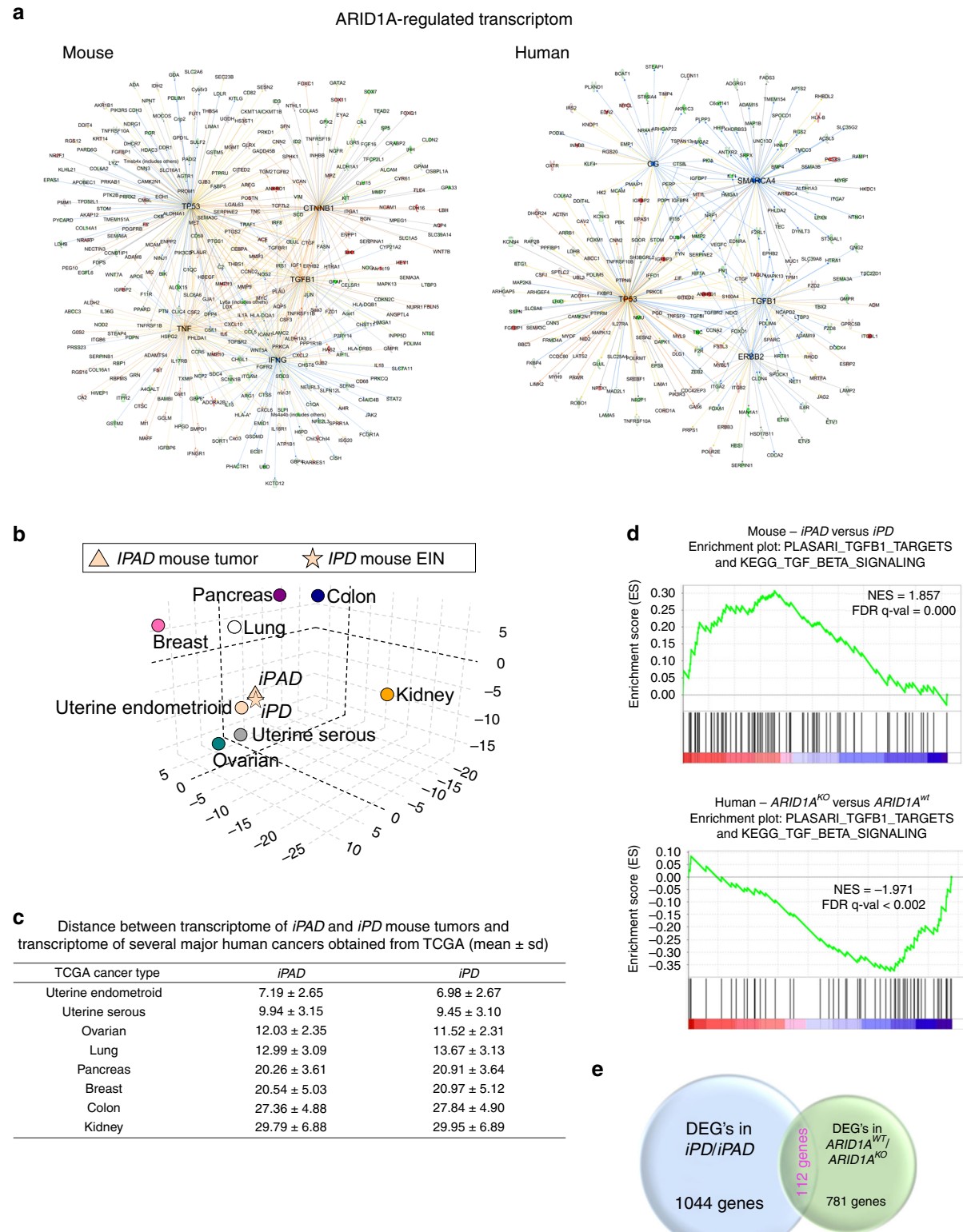

**a** ARID1A-regulated transcriptom

Mouse — Human

**b** △ *IPAD* mouse tumor ☆ *IPD* mouse EIN

**c** Distance between transcriptome of *iPAD* and *iPD* mouse tumors and transcriptome of several major human cancers obtained from TCGA (mean ± sd)

| TCGA cancer type | iPAD | iPD |
|---|---|---|
| Uterine endometroid | 7.19 ± 2.65 | 6.98 ± 2.67 |
| Uterine serous | 9.94 ± 3.15 | 9.45 ± 3.10 |
| Ovarian | 12.03 ± 2.35 | 11.52 ± 2.31 |
| Lung | 12.99 ± 3.09 | 13.67 ± 3.13 |
| Pancreas | 20.26 ± 3.61 | 20.91 ± 3.64 |
| Breast | 20.54 ± 5.03 | 20.97 ± 5.12 |
| Colon | 27.36 ± 4.88 | 27.84 ± 4.90 |
| Kidney | 29.79 ± 6.88 | 29.95 ± 6.89 |

**d** Mouse – *iPAD* versus *iPD*
Enrichment plot: PLASARI_TGFB1_TARGETS and KEGG_TGF_BETA_SIGNALING

NES = 1.857
FDR q-val = 0.000

Human – *ARID1A^{KO}* versus *ARID1A^{wt}*
Enrichment plot: PLASARI_TGFB1_TARGETS and KEGG_TGF_BETA_SIGNALING

NES = −1.971
FDR q-val < 0.002

**e** DEG's in *iPD*/*iPAD* — 1044 genes | 112 genes | DEG's in *ARID1A^{WT}*/*ARID1A^{KO}* — 781 genes

whether ARID1A pathway causally affects TGF-β expression and its related phenotypes. TGF-β is a multifunctional growth factor, and depending on the context, it participates in proliferation, differentiation, migration, and immune suppression/evasion[27,28]. TGF-β signaling is triggered by ligand (such as TGF-β1) binding to heterotetrameric receptors (such as TGFBR2) located on the cytoplasmic membrane, resulting in phosphorylation events that activate signal transducers including SMADs in the nucleus.

To determine if *Arid1a* inactivation attenuated TGF-β signaling in murine models, we performed immunohistochemistry of phospho-Smad3 (Ser465/467), a downstream target of TGF-β, to assess TGF-β signaling activity. Phospho-Smad3 immunoreactivity in uterine epithelial cells was found to be weaker in *iAD* mice with induced *Arid1a*-deletion than in mice without induced knockout (Fig. 4a and Supplementary Fig. 4b). We also measured pSMAD3 and total SMAD3 levels by western blot in a pair of

**Fig. 3 Network and distance analysis of the transcriptome of *iPAD* and *iPD* mouse endometrial tumors. a** Interconnected networks of the top six upstream regulators predicted by IPA from the ARID1A-regulated transcriptome in mouse (left) and in human (right). Magenta color indicates transcripts upregulated and green color indicates transcripts downregulated following *ARID1A* deletion. **b** Euclidean molecular distance matrix assessing the similarity between transcriptome derived from *iPAD* mouse uterine tumors, *iPD* mouse endometrioid intraepithelial neoplasis (EIN), and TCGA transcriptome data obtained from several major types of human cancers. **c** Euclidean distance between the transcriptome data of mouse endometrial tumors and 7 major types of human cancer from TCGA. The values show the mean and standard deviation of the distances in 3-D principal component analysis space. **d** Enrichment plots from gene set enrichment analysis (GSEA). GSEA was performed to assess enrichment of TGF-β target gene signature[23] in ARID1A-associated mouse (FDR = 0.000) and human (FDR < 0.002) transcriptomes. NES: normalized enrichment score. FDR values were determined by Benjamini–Hochberg FDR correction. **e** Venn diagrams of differentially expressed genes (DEGs) in mouse and human transcriptomes. The DEGs between *iPAD* and *iPD* mouse tumors and the isogenic pairs of *ARID1A^{WT}* and *ARID1A^{KO}* cells were identified in Supplementary Data 1 and Supplementary Data 3, respectively. The numbers of DEGs in each group and the overlap between groups are shown in the Venn diagrams.

isogenic *ARID1A^{KO}* and *ARID1A^{WT}* cells and observed a reduction of intrinsic pSMAD3 levels in *ARID1A^{KO}* cells compared to *ARID1A^{WT}* cells (Fig. 4b, lanes 1 and 2). *ARID1A^{WT}* cells responded to the extracellular ligand TGF-β1, as evidenced by enhanced pSMAD3 levels (Fig. 4b, compare lanes 1 and 3). In contrast, the TGF-β1 ligand-induced response is less pronounced in *ARID1A^{KO}* cells (Fig. 4b, compare lanes 2 and 4). After normalizing pSMAD3 to total SMAD3, the quantitative ratio of pSMAD3 in the presence or absence of TGF-β1 was 4.6 fold for *ARID1A^{KO}* cells and 7.3 fold for *ARID1A^{WT}* cells (Fig. 4c). Additionally, in comparison to *ARID1A^{WT}* cells, *ARID1A^{KO}* cells showed a diminished response to TGF-β1, as demonstrated by a reduced pSMAD3/SMAD3 ratio (Fig. 4c, $p < 0.001$, repeated measures ANOVA). Collectively, data from both the in vivo tumor models and cell culture systems indicate that *ARID1A* loss resulted in attenuated endogenous TGF-β signaling activities and reduced TGF-β response.

Since the TGF-β pathway can be involved in cellular growth, movement, and invasion[27], we determined if *ARID1A* knockout in endometrial epithelial cells affected these tumor-promoting phenotypes. First, we measured cellular proliferation and viability, but did not observe a significant difference between *ARID1A^{WT}* and *ARID1A^{KO}* cells (Fig. 4d). Cell motility was measured by the rate of wound closure. We found that *ARID1A^{KO}* cells (light pink line, Fig. 4e) migrated and closed the wound gap faster than *ARID1A^{WT}* cells (light green line, Fig. 4e). Interestingly, *ARID1A^{WT}* cells responded to the inhibitory effect of extracellular TGF-β1 (Fig. 4e, dark vs. light green lines). In contrast, the migration rate of *ARID1A^{KO}* cells was not affected by the presence of TGF-β1 (Fig. 4e, compare magenta vs. light pink lines). Cellular invasion through the extracellular matrix was assessed by real-time electrical impedance assay using xCELLigence with Matrigel-coated microelectrodes. Tumor cells capable of dissolving and invading through the Matrigel layer and attaching to the microelectrodes caused changes in impedance, which directly correlated with their invasive capacity. In this assay, *ARID1A^{KO}* cells were more invasive than *ARID1A^{WT}* cells (Fig. 4f, light pink vs. light green lines). Invasive potency of *ARID1A^{WT}* cells was suppressed by exogenous TGF-β1, while *ARID1A^{KO}* cells were not (Fig. 4f). On the other hand, transient re-expression of ectopic ARID1A in *ARID1A^{KO}* cells restored their sensitivity to TGF-β1, as shown by our data that upon exposure to TGF-β1, there was a clear reduction in invasive capacity of these cells (Supplementary Fig. 4C).

We then used SB431542 (1 and 10 μM) to inhibit endogenous TGF-β signaling. Our result from a single experiment indicated that the addition of SB431542 showed a trend of increasing wound closure rate in *ARID1A^{WT}* cells (Supplementary Fig. 4D, $p = 0.097$, slope comparison–linear regression test), while the closure rate of *ARID1A^{KO}* cells was not affected (Supplementary Fig. 4D, $p = 0.983$, slope comparison–linear regression test).

Importantly, using a more sophisticated xCELLigence invasion assay, we observed that *ARID1A^{WT}* cells had enhanced invasive capacity at both 1 and 10 μM, while *ARID1A^{KO}* had a significant effect only at 10 μM but not at 1 μM (Supplementary Fig. 4E). SB431542 treatment in *ARID1A^{WT}* also enhanced the cell invasion rates, similar to that in *ARID1A^{KO}* without SB431542 treatment, as there was no significant difference between these two groups (Supplementary Fig. 4E). Since cell growth rate was not different between *ARID1A^{WT}* and *ARID1A^{KO}* cells (Fig. 4d), the observed differences in TGF-β1 response are most likely reflective of the functional role of ARID1A.

Cell migration patterns monitored by live cell tracking were random for both cell lines in the absence of the TGF-β1 gradient (Fig. 4g, top panels). In the presence of a TGF-β1 gradient, *ARID1A^{WT}* cells displayed repulsive trajectories whereas *ARID1A^{KO}* cells continued to migrate randomly (Fig. 4g, bottom panels). Quantitatively, there was no intrinsic difference in cell forward migration index (xFMI) between *ARID1A^{WT}* and *ARID1A^{KO}* in the absence of TGF-β1 (Fig. 4h). However, the presence of extracellular TGF-β1 induced a significant xFMI difference between *ARID1A^{WT}* and *ARID1A^{KO}* cells (Fig. 4h).

**ARID1A binds at regulatory regions to control gene transcription.** By analyzing the ARID1A-regulated transcriptome, we found several members of the TGF-β signaling pathway including *TGFBR2*, *TGFBI*, and *BMP4* were significantly downregulated in *ARID1A^{KO}* cells (Supplementary Fig. 4F). We then validated the ARID1A-regulated transcription of *TGFBR2* in three different cell line models by qRT-PCR (Supplementary Fig. 5A). Suppression of *TGFBR2* was evident in both human endometrial epithelial and HCT116 human colorectal carcinoma *ARID1A^{KO}* cells, while *TGFBR2* was upregulated in OV207 cells with induced *ARID1A* expression.

To address whether the ARID1A chromatin remodeler bound to their transcriptional regulatory regions, a prerequisite step of transcriptional control, we performed genome-wide ChIP-seq using antibodies specific to ARID1A. We also performed ChIP-seq on BRG1, a protein forming the SWI/SNF complex with ARID1A[29], to facilitate accurate identification of the ARID1A-containing SWI/SNF complex binding sites. To assess whether ARID1A-containing SWI/SNF complex affected chromatin accessibility and recruitment by RNA polymerase II (RNA Pol II), we performed ATAC-seq and RNA Pol II ChIP-seq. All of the ChIP-seq and the ATAC-seq studies were performed using two biological replicates of the isogenic pairs of human endometrial epithelial cell lines.

The ChIP-seq experiments were conducted according to ENCODE guidelines (version 3.0) and revealed 41,292 high-confidence ARID1A consensus ChIP-seq peaks in *ARID1A^{WT}* cells that were absent from *ARID1A^{KO}* cells (Supplementary Fig. 5B; FDR < 0.01, Benjamini–Hochberg FDR correction). ARID1A peaks primarily localized within intron (42.1%),

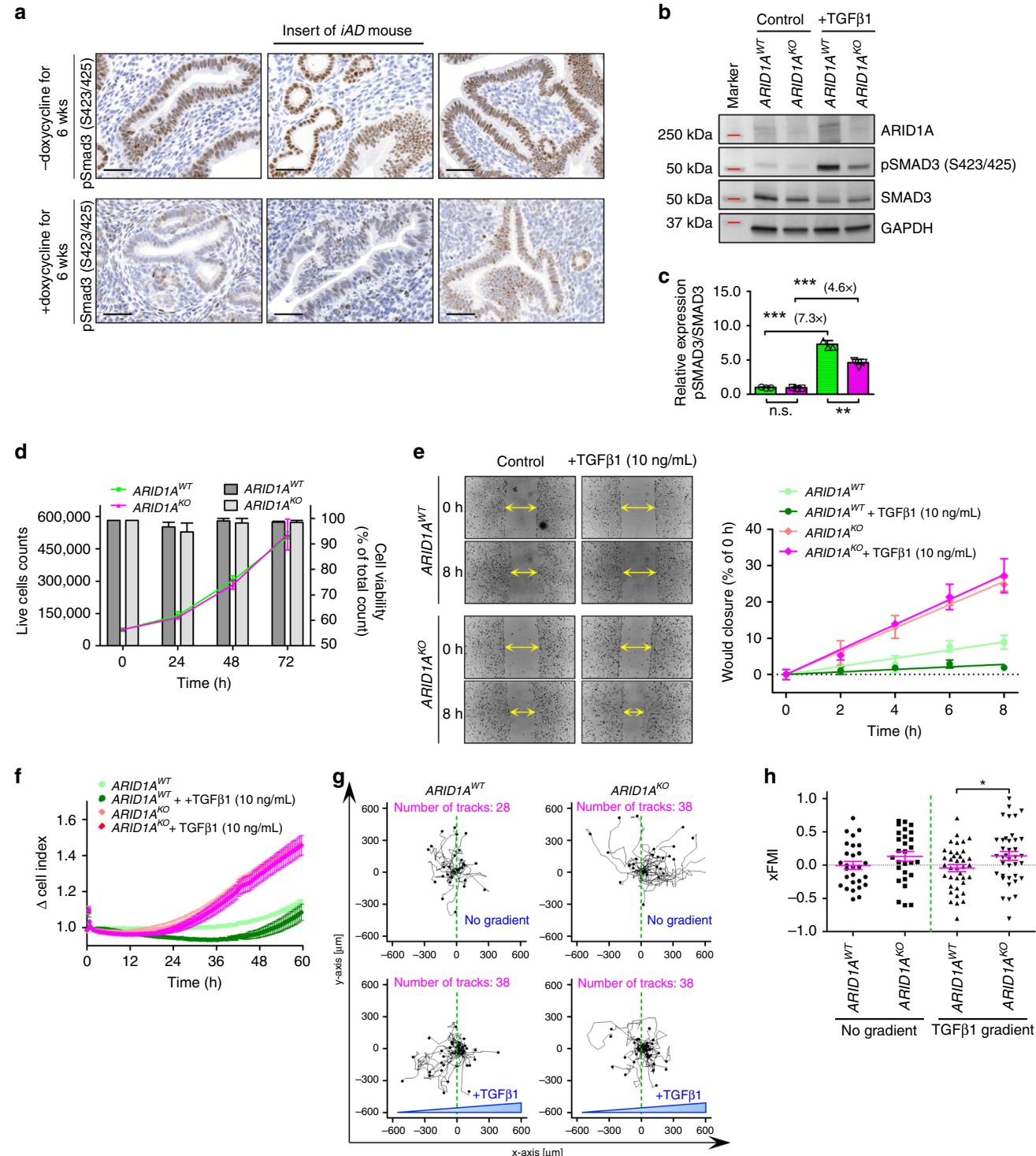

intergenic (28.0%), enhancer (17.5%), and promoter (7.24%) regions (Supplementary Fig. 5C). Heatmap profiling of the ARID1A ChIP-seq peaks and surrounding regions demonstrated the specificity of ARID1A binding events, with minimal background peaks observed in $ARID1A^{KO}$ cells (Fig. 5a). Quality of all ChIP-seq experiments was summarized in Supplementary Data 6. We then used HOMER software package to identify DNA binding motifs enriched in the ARID1A ChIP-seq target peaks and found that the DNA binding motifs for JUNB, TEAD4, RUNX1, EWS:ERG, and NF1 were significantly enriched (Supplementary Fig. 5D).

BRG1 ChIP-seq performed on $ARID1A^{WT}$ and $ARID1A^{KO}$ cell lines demonstrated a 40% reduction of BRG1 binding events in the $ARID1A^{KO}$ cells (Fig. 5a and top Venn diagram on Supplementary Fig. 5e). A significant association between ARID1A and BRG1 ChIP-seq binding events was observed in $ARID1A^{WT}$ cells. Based on Pearson correlation on ChIP-seq data, we found that ARID1A- and BRG1- ChIP-seq data from $ARID1A^{WT}$ cells were the most similar group among all the comparisons groups (Fig. 5b). Co-localization coefficient as shown in Fig. 5c indicated a high co-binding between ARID1A and BRG1 ChIP-seq on $ARID1A^{WT}$ cells (Fig. 5c, co-binding

**Fig. 4 Loss of *ARID1A* attenuates response to inhibitory TGF signal resulting in enhanced cell motility and invasion. a** Representative pSmad3 (Ser465/467) immunohistochemical stains of *iAD* mouse uterine tissues. Staining shows the effect of Arid1a protein loss on pSmad3 (Ser465/467) in doxycycline-treated *iAD* mice. Mice were treated with doxycycline for 6 weeks. Low magnification images are shown in Supplementary Fig. 4B. Scale bars, 50 μm. **b** Representative immunoblots showing pSMAD3 (S465/467), total SMAD3, and ARID1A protein expression in *ARID1A^KO^* and *ARID1A^WT^* cells. GAPDH was used as a loading control. **c** Densitometric quantification of immunoblot images shown in **b** showing that *ARID1A^KO^* cells lost responsiveness to TGF-β1 in comparison to *ARID1A^WT^* cells. Data show the ratio of phospho-SMAD3 to total SMAD3. Data are presented as mean ± SD ($n = 3$); **$p < 0.01$; ***$p < 0.001$; n.s., not significant; repeated measures ANOVA. Green, *ARID1A^WT^*; magenta, *ARID1A^KO^*; green dotted, *ARID1A^WT^* + TGF-β1; magenta dotted, *ARID1A^KO^* + TGF-β1. **d** Measurement of cellular proliferation and cell viability for *ARID1A^WT^* and *ARID1A^KO^* cells over 72 h. Line graphs show cellular proliferation (left *y*-axis); *ARID1A^WT^* cells (green), *ARID1A^KO^* cells (magenta). Cellular viability (right *y*-axis) is shown as bar graphs; *ARID1A^WT^* cells (dark gray), *ARID1A^KO^* cells (light gray). Data are expressed as mean ± SEM ($n = 3$). **e** Representative images of the wound-healing assay performed on *ARID1A^WT^* and *ARID1A^KO^* isogenic cells in the absence or presence of TGF-β1 (left panel). The effects were quantified by the distance of the gaps in the wounds (right panel). Yellow arrows indicate the size of the wound. Mean ± SEM ($n = 3$) is shown in the quantification graph. **f** Real-time cell impedance assay using the xCELLigence real-time cell analysis system in *ARID1A^WT^* and *ARID1A^KO^* cells. The capability of cells to invade through the Matrigel matrix barrier to the lower chamber in the absence or presence of TGF-β1 gradient is presented as delta cell index (vertical axis). Mean ± SEM ($n = 3$) is shown. **g** Live-cell trajectory tracings showing two-dimensional movement tracks of individual *ARID1A^WT^* and *ARID1A^KO^* cells in the absence (top panel) or presence (bottom panel) of TGF-β1 gradient. **h** Scatter plot of the forward migration index in the x-direction (xFMI) of the two-dimensional movement measured in **g**. Red bars represent mean ± SEM. *$p < 0.05$; paired two-tailed Student *t*-test.

coefficient = 0.99). On the other hand, BRG1 ChIP-seq binding events at ARID1A ChIP-seq target regions were decreased in *ARID1A^KO^* cells (Fig. 5b, c). The data suggested that the presence of ARID1A was critical in directing or affecting the chromatin binding and localization of BRG1.

While there is a modest co-localization between ARID1A ChIP-seq peaks and RNA Pol II ChIP-seq binding events (co-localization coefficient = 0.36), the co-localization between ARID1A ChIP-seq peaks and ATAC-seq events was strong (co-localization coefficient = 0.65) (Fig. 5c, top). Comparing ATAC-seq signals, we identified 15,192 (38.6%) ATAC-seq peaks were lost in concordance to ARID1A knockout status, of which 8,979 peaks were associated with ARID1A-specific binding events (Supplementary Fig. 5f). We also found that 496 ATAC-seq signals were gained in response to *ARID1A* loss, of which 33 peaks were associated with ARID1A-specific peaks (Supplementary Fig. 5F). Moreover, loss of ATAC-seq signals was more intensely associated with differential gene expression following *ARID1A* knockout than the ATAC-signals gain (hypergeometric *p*-value of 6.60E−36 vs 5.56E−5, respectively).

Using SWI/SNF ChIP target peaks to designate co-binding events of ARID1A and BRG1 specific to *ARID1A^WT^* cells (approach shown in Supplementary Fig. 5E), a strong correlation between the SWI/SNF ChIP target peaks and ATAC-seq peaks could be observed in *ARID1A^WT^* cells, but not in *ARID1A^KO^* cells (Fig. 5c; co-localization coefficient 0.60 vs. 0.10). Together, the data suggested that chromatin accessibility was highly likely to be co-regulated by ARID1A and BRG1. To identify direct target genes of the ARID1A chromatin remodeling complex, we integrated ARID1A-BRG1 ChIP co-binding targets with genes showing significant differential expression (FDR < 0.05, Benjamini–Hochberg FDR correction, and fold change >1.4) when *ARID1A* was knocked out (Fig. 5d). This integrative analysis identified 567 genes (Fig. 5d and Supplementary Data 7), including *THBS1*, a well-known ARID1A target gene *THBS1*[30] was detected, as well as genes in the TGF-β signaling pathway including *TGFBI*, *TGFBR2*, *THBS1*, *BMP4*, *ID4*, and *TNC* (Supplementary Data 8 and Supplementary Fig. 4F). To further delineate the disease relevance of these ARID1A target genes, we assessed their expression in TCGA uterine endometrial carcinoma PanCancer Atlas dataset. Examples of correlation between ARID1A and its target genes in the TGF-β signaling pathway, *ID4*, *KLF10*, *TGFBR2*, and *TRPS1*, are shown in Supplementary Fig. 5G.

Ingenuity pathway analysis on ARID1A direct target genes further uncovered enrichment of VEGF, IGF-1, and IL-8 signaling pathways, indicating they are potential functional mediators of ARID1A (Supplementary Data 9). ChIP-seq and ATAC-seq peaks of the representative TGF-β genes were presented in Fig. 5e. For most of the genes, deletion of *ARID1A* as occurred in *ARID1A^KO^* cells significantly affected chromatin accessibility as well as binding of BRG1 to transcriptional regulatory regions, further supporting a direct regulatory role of ARID1A-BRG1 complex in transcription.

## Discussion

Uterine endometrioid carcinoma is the most common gynecologic cancer. In this study, we investigated the role that *ARID1A* inactivating mutations in the endometrium might play in the development and progression of endometrioid carcinoma. We found that conditional deletion of *Arid1a* alone in endometrial epithelial cells was insufficient to induce neoplastic transformation, whereas conditional deletion of *Pten* led to endometrial hyperplasia that progressed to carcinoma without myometrial invasion. Interestingly, *Arid1a* deletion coupled with *Pten* deletion remarkably accelerated tumor progression by increasing invasiveness and dissemination to the lymph nodes 6 weeks after gene knockout. This rapid progression was associated with a significantly shorter life span than seen in *Pten*-deleted (*iPD*) mice. Histopathological, clinical, and molecular features of tumors from *Arid1a/Pten* co-deleted (*iPAD*) mice closely resembled human uterine endometrioid carcinomas. Our data also indicated that deleterious mutations in *ARID1A* promote tumor invasion and metastasis. This function is probably mediated by aberrant TGF-β signaling and was not mechanistically validated previously. Finally, the mouse models created in this study are valuable and highly relevant for testing new cancer therapies including immune-based intervention as these mice models maintain immunocompetency.

The ARID1A-regulated transcriptome identified in mouse endometrial tumors and human endometrial epithelial cells demonstrated TGF-β and TP53 as the two important signaling networks mediated by ARID1A. A direct link between ARID1A and p53 signaling has been reported previously by our group and others which highlighted the regulation of cell cycle and modulation of p21 levels by ARID1A[5,31,32]. The discovery that key components of the TGF-β signaling pathway, including the TGFBR2, are direct targets of ARID1A illustrates the pleiotropic function of ARID1A. The high confidence ARID1A-target genes identified by integrating ARID1A/BRG1 ChIP-seq peaks with the ARID1A-regulated transcriptome further support this view. Several functional pathways critical for endometrial

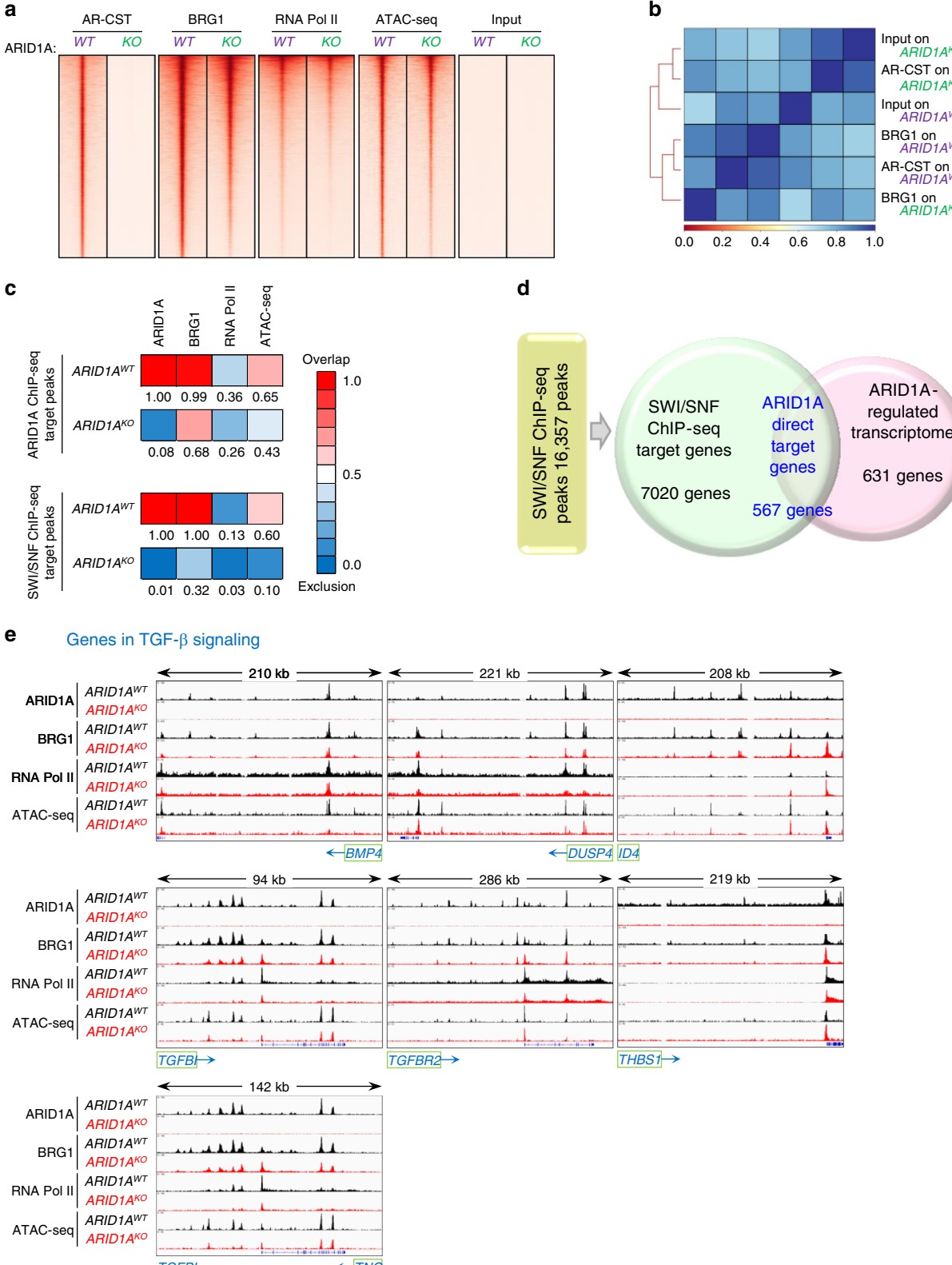

tumorigenesis are also enriched in ARID1A-target genes include VEGF, IL-8, Stem Cell Pluripotency, and TGF-β (Supplementary Data 9). Thus, the observed synergy between *ARID1A* and *PTEN* deletion in endometrial tumorigenesis cannot be solely attributed to the alteration in TGF-β signaling. Aberrations in other functional pathways resulting from *ARID1A* inactivating mutations

may well account for the observed aggressive phenotype in the *iPAD* mouse model and human endometrioid carcinomas.

Our functional analysis showed that *ARID1A*-inactivation releases the brake on cell migration and invasion imposed by TGF-β signaling. In agreement with our observations, an independent study of a *Tgfbr1* and *Pten* double conditional knockout

**Fig. 5 Integrated analysis of ChIP-seq and RNA-seq data from isogenic *ARID1A^{WT}* and *ARID1A^{KO}* cells. a** Heatmaps of the ARID1A, BRG1, and RNA polymerase II ChIP-seq peaks as well as ATAC-seq peaks in *ARID1A^{WT}* (*WT*) and *ARID1A^{KO}* (*KO*) cells. Each peak is represented as a horizontal line, ordered vertically by signal strength. ARID1A ChIP-seq peak signals were minimal in *ARID1A^{KO}* cells, demonstrating the specificity of the antibody and the experimental conditions. **b** Co-localization coefficient matrix of ARID1A and BRG1 ChIP-seq peaks in *ARID1A^{WT}* and *ARID1A^{KO}* cells. The strength of correlation is presented by color gradient. Dark blue to dark red: highest to no correlation. AR-CST, ARID1A antibody from Cell Signaling Technology. **c** Heatmaps showing overlapping frequencies of ARID1A ChIP-seq target peaks (top) and SWI/SNF ChIP-seq target peaks (bottom) with ChIP-seq peaks from various target proteins. The rightmost box (ATAC-seq) represents the number of co-localization events between accessible chromatin regions identified by ATAC-seq and ARID1A ChIP-seq target peaks. The strength of peak co-localization is depicted by color gradient with dark red representing overlapping peaks and dark blue representing disjoint peaks. **d** Venn diagram showing ARID1A directly regulated genes by integrated SWI/SNF ChIP-seq and ARID1A-regulated transcriptome analyzed in human endometrial epithelial cells. A total of 594 ARID1A direct target genes were identified. **e** UCSC genome browser view of ARID1A. BRG1, and RNA polymerase II occupancy and ATAC-seq event at the locus of representative ARID1A target genes. Each track represents normalized ChIP-seq (or ATAC-seq) events as described in the methods. *ARID1A^{KO}* cells were used as a control for each factor. ChIP-seq binding events and ATAC-seq tracings at the locus of representative ARID1A target genes. Blue arrow indicates the transcriptional orientation of each gene.

mouse exhibited rapidly progressing endometrial tumors that exhibited myometrial invasion, which remarkably resembled the phenotypes observed in *iPAD* mice[17]. Taken together, these data suggest that ARID1A may affect the TGF-β axis, which in turn contributes to tumor cell infiltration into and through the myometrial wall, a critical step in metastasis. This understanding of the molecular functions of ARID1A, including its contribution to increased risk of disease progression and metastasis, is expected to impact future translational studies in prognostic and therapeutic endeavors.

Although our results have illuminated additional mechanistic role of ARID1A in endometrial tumorigenesis, several questions remain to be answered. It is important to determine whether the epigenomic landscape is affected by alterations in the ARID1A SWI/SNF complex at different stages of endometrial tumor development or in response to endogenous and exogenous factors such as hormones, obesity, and exposure to infection or carcinogen. Additionally, the mechanism of synergistic effects of *PTEN* and *ARID1A* inactivation is unclear. *PTEN* transcription has been reported to be directly and positively regulated by SMAD3, a downstream effector molecule of TGF-β signaling[33]. *ARID1A* loss may lead to the downregulation of TGFBR1/2, which causes attenuated SMAD3 phosphorylation and subsequently affecting *PTEN* transcription. This would result in enhanced PI3K/AKT signaling, increased proliferation and resistance to apoptosis[33]. This observation lends further support to the cooperative interactions between ARID1A/TGF-β axis and PTEN in the pathogenesis of endometrioid carcinoma of the uterus. Since the TGF-β pathway affects local immunity, and the enrichment of pro-inflammatory chemokines, Tnf, Il-6, and Il-17, is identified in *iPAD* tumors (Supplementary Data 2), future studies are needed to assess the functional roles of ARID1A in regulating host immunity. Thus, the establishment of a syngeneic, immunocompetent *iPAD* mouse model opens avenues for these endeavors. In summary, the findings from this study provide molecular insights into how *ARID1A* inactivation modulates transcription reprogramming to accelerate endometrial tumor progression and dissemination, which are the major causes of cancer mortality.

## Methods

**Cell culture**. The immortalized parental human endometrial epithelial and the first pair of isogenic *ARID1A^{WT}* and *ARID1A^{KO}* cells used in this study have been reported[20,24]. Anonymous endometrial tissue was obtained from pre-menopausal women who underwent hysterectomy for a benign reason. Acquisition of tissue specimens was approved by the Institutional Review Board at the Johns Hopkins Hospital, Baltimore, Maryland. Fresh samples were directly processed after arrival in the surgical pathology unit. Endometrial tissue was rinsed with phosphate-buffered saline (Life Technologies), and minced thoroughly in a petri dish. The cells were further dissociated using 5 ml of 0.05% trypsin (Life Technologies).

Trypsin activity was later neutralized using 10 ml RPMI1640 medium (Life Technologies) containing 10% FBS (Sigma). Primary cultures were maintained in RPMI1640 medium supplemented with 15% FBS, 1% penicillin/streptomycin (Life Technologies), and 1% non-essential amino acids (Life Technologies). Cells were seeded on 0.1% gelatin-coated dishes at 37 °C and 5% $CO_2$. Epithelial cells were immortalized using SV40 adenovirus and were designated as human endometrial epithelial cell line. These cells expressed epithelial cell markers including cytokeratin-8, Ep-CAM, and E-cadherin, as well as estrogen receptor. More than 99% of these cells were positive for Ep-CAM assessed by flow cytometry. This cell line harbored wild-type ARID1A and expressed BAF250. The human endometrial epithelial cells were authenticated using STR DNA analyses as being of human origin, with no match (cutoff = 80%) to any reference profile in the ATCC or DSMZ database.

A second isogenic line of human endometrial epithelial cells with *ARID1A* knockout were generated in the current study. CRISPR/Cas9 pSpCas9n(BB)−2A-Puro (PX462) vector from Addgene (plasmid #48141) was used in this study. Cloning was performed using a pair of CRISPR single-guide RNA (sgRNA) specifically targeting exon 15 of ARID1A[24], namely top strand-nicking sgRNA AACGGCGGGATGGGTGACCC and bottom strand-nicking sgRNA TACAGTCGTGCTGCCGGCCC. Resulting plasmid were transfected into cells using Lipofectamine® 3000 (Life Technologies), and positive cells were selected in the presence of 1 μg/ml puromycin. The efficiency of ARID1A knock-out was verified by sequencing and Western blotting. The *ARID1A^{KO}* cells were confirmed to be 100% isogenic to the respective *ARID1A^{WT}* parental cells. All cell lines tested negative for mycoplasma. STR and mycoplasma testing were carried out by Johns Hopkins GRCF DNA Services/FAF facility.

**Mice**. *Arid1a^{flox/flox}* mice on the 129S1 background and *Pten^{flox/flox}* mice on the BALB/c background (Strain C;129S4-Pten^{tm1Hwu}/J) were obtained from the Jackson Laboratory. Cre-mediated deletion of *Arid1a* removes exon 8 and introduces a frameshift mutation and a premature stop codon (p.Gly809Hisfs*6)[31]. Cre-mediated deletion of *Pten* removes exon 5 and introduces a frameshift mutation (p.Val85Glyfs*14)[34]. *Arid1a^{flox/flox};Pten^{flox/flox}* mice were generated by intercrossing these transgenic strains. In order to express Cre recombinase specifically in the mouse uterine epithelium for all subsequent genetic alterations described, we used Pax8-Cre mice which were generated by crossing mice expressing the reverse tetracycline-controlled transactivator (rtTA) under the control of the Pax8 promoter (Pax8-rtTA) with mice expressing Cre recombinase in a tetracycline-dependent manner (TetO-Cre)[19].

To generate mouse models with *Arid1a* and *Pten* individual or combined knockout in the uterine epithelium, we crossed *Pax8-Cre* mice with *Arid1a^{flox/flox}* mice, *Pten^{flox/flox}* mice, and *Arid1a^{flox/flox};Pten^{flox/flox}* mice. All experimental mice were maintained on a mixed genetic background (C57BL/6, BALB/c, and S129). The resultant genotype of each model was confirmed by genomic DNA PCR, using primers listed in Supplementary Data 10. Knockout was initiated by treating mice with doxycycline either through oral gavage (2 mg/mouse/day) or subcutaneous implantation of doxycycline pellets (200 mg) when they reached puberty (6–8 weeks old). All of the animal procedures were approved by the Johns Hopkins University Animal Care Committee.

**Chromatin immunoprecipitation assay**. The chromatin immunoprecipitation (ChIP) assay was performed on the two independent clones of human endometrial epithelial *ARID1A^{WT}* and *ARID1A^{KO}* cells[24] with modifications. Cells were grown in 15 cm dishes, and ~1.2 × 10^7 cells were cross-linked using Diagenode ChIP cross-link Gold and 1% formaldehyde according to the manufacturer's instruction. Nuclear contents were extracted using the truChIP Chromatin Shearing kit (Covaris) according to the manufacturer's instructions. Chromatin was sheared for 12 min in shearing buffer using a Covaris E220 focused ultrasonicator. The

fragment sizes were ensured to be between 200 and 600 bp. The sonicated lysates were diluted 5-fold with ChIP dilution buffer (0.1% Triton X-100, 2 mM EDTA, 20 mM Tris-HCl pH 7.5, 150 mM NaCl, and 1× protease inhibitor), and immunoprecipitated with rotation overnight at 4 °C with 0.5–3 μg of antibodies. The antibody/chromatin complex was then precipitated for 3 h by Protein A/G DYNAL magnetic beads (40 μl of 1:1 mixture). Antibody-protein complexes bound to beads were washed once with low salt buffer (20 mM Tris-HCl pH 7.5, 150 mM NaCl, 0.1% SDS, 1% Triton-X100, and 2 mM EDTA), once with high salt buffer (20 mM Tris-HCl pH 7.5, 500 mM NaCl, 0.1% SDS, 1% Triton-X100, and 2 mM EDTA), once with LiCl buffer (250 mM LiCl, 1% NP-40, 1% sodium deoxycholate, 1 mM EDTA, and 10 mM Tris-HCl pH 8.0), and twice with TE, pH 8.0. DNA and protein complexes were digested in TE buffer containing 1% SDS, 200 mM NaCl, and 1 U Proteinase K (Thermo Scientific) at 56 °C for 2 h, and cross-linking was reversed by heating at 65 °C for 4 h. DNA fragments were purified using a QIAquick PCR Purification Kit in 55 μl of EB elution buffer. Tru-seq ChIP-seq library preparation and sequencing using a NextSeq500 platform with single-end reads of 75 bases were performed by JHMI Deep Sequencing and Microarray Core.

**Chromatin accessibility assay.** ATAC-seq library construction[35] was performed using 60,000 cells of two independent clones of human endometrial epithelial $ARID1A^{WT}$ and $ARID1A^{KO}$ cells. Cells were washed in PBS, pelleted by centrifugation, and lysed 50 μl cold lysis buffer (10 mM Tris-Cl, pH 7.4, 10 mM NaCl, 3 mM $MgCl_2$, and 0.1% IGEPAL CA-630). Crude nuclei isolates were incubated in 50 μl tagmentation mix (1× TD buffer, 2.5 μl Tn5, Illumina) at 37 °C for 30 min. Tagmented DNA was purified using a QIAgen MinElute PCR Purification kit, and was subsequently amplified using NEBNext High Fidelity 2× PCR master mix for 11 cycles. PCR products were purified using Agencourt AMPure XP (Beckman Coulter), and sequencing was performed by JHMI Deep Sequencing and Microarray Core using a NextSeq500 platform with double-end reads of 75 bases.

**RNA-seq and analysis.** Uteri from 4 iPAD and 4 iPD mice were harvested two weeks after gene knockout (by feeding doxycycline). Hematoxylin-and-eosin-stained tissue sections were prepared, and the diagnosis of endometrioid carcinoma from all iPAD mice and EIN from all iPD mice was confirmed by histopathology. Two isogenic lines of human endometrial epithelial cells with ARID1A knockout were grown under normal growth- and serum starved (normal growth media minus FBS)-conditions for 24 h. Total RNA was isolated from cultured human endometrial epithelial cells and mouse uterine tissues using the Qiagen RNeasy Plus Mini Kit. The RNA quality was assessed using the Agilent 2100 Bioanalyzer RNA Nano Chip. RNA-sequencing was performed by GeneWiz, Inc. on the Illumina HiSeq2500 platform, in a 2 × 100 bp paired-end high output V4 chemistry configuration.

Bioinformatics analysis was performed using Galaxy, an open access web-based program that contains a variety of next-generation sequencing analysis tools. Reads were processed and aligned to *Homo sapiens* reference genome build hg19 or *Mus musculus* reference genome build mm10 using TopHat gapped-read mapper (ver. 2.1.0). The aligned reads were processed with Cufflinks transcript assembly (ver. 2.2.1.0), and output of the resulting GTF files was amalgamated to UCSC hg19 or mm10 RefSeq genes annotation files using Cuffmerge (ver. 2.2.1.0). Differential expression analysis was performed using Cuffdiff (ver. 2.2.1.3) to produce a list of genes whose expression changes were significant (FDR < 0.05, Benjamini–Hochberg FDR correction) between the groups tested.

Principal component analysis was applied to the data to identify the first three principal components carrying the most substantial variances. A 3D scatter plot in the dimensions of three principal components was generated to show the distance between mouse samples and the TCGA cancer types. Data points were created according to the mean values of samples of the corresponding cancer type. Pairwise Euclidean distance between the mouse and the TCGA samples from different cancer types was then calculated.

**cDNA synthesis and quantitative RT-PCR.** First strand cDNA was generated using the iScript cDNA synthesis kit (Bio-Rad). Quantitative RT-PCR was performed using OneTaq® Hot Start DNA polymerase (New England Biolabs) and SYBR Green I (Life Technologies). The primers used for Quantitative RT-PCR are listed in Supplementary Data 10.

**Western blotting.** Protein extraction was carried out using Cell Lysis Buffer (Cell Signaling Technologies) supplemented with Halt Protease inhibitor cocktail (Thermo Scientific) and PhosSTOP (Roche). Proteins were separated using 4–15% Mini-PROTEAN® TGX™ Precast Protein Gels (Bio- Rad), followed by transfer to 0.2 μm PVDF, and immunoblot analyses were performed. Blots were developed using the Clarity™ Western ECL Blotting Substrate.

**Histology and immunohistochemistry.** All tissues were fixed with 10% neutral buffered formalin for 24 to 48 h. Tissues were processed by The Johns Hopkins University Oncology Tissue Services using a standard tissue processing protocol, embedded in paraffin, and 5 μm sections were cut. Hematoxylin-and-eosin-stained sections were prepared from uteri, fallopian tube, and ovarian tissues from all mice after necropsy. We classified endometrial lesions by applying diagnostic criteria

similar to those used in women, cognizant of the fact that tumors from mouse and human are different. In clinical pathology diagnosis, low-grade (grade 1) endometrial carcinoma is characterized by low-grade nuclei with architectural solid component less than 5%, grade 2 with solid component between 5% to 50% and grade 3 with more than 50% or grade 1 and grade 2 with high-grade nuclei.

Formalin fixed paraffin-embedded sections were deparaffinized and rehydrated. Antigen retrieval was carried out using DAKO Target Retrieval Solution, equivalent to citrate buffer pH 6.0, or Trilogy, equivalent to neutral pH EDTA. Endogenous peroxidase activity was quenched in 3% $H_2O_2$. Sections were incubated with antibodies overnight at 4 °C. Immunostained sections were visualized using DAKO EnVision+ System-HRP goat Anti-Rabbit IgG and DAKO DAB+ Substrate Chromogen System. Nuclei were visualized using hematoxylin counterstaining. Cover slides were mounted with Cytoseal 60. We used commercial antibodies that have been validated and/or published for immunohistochemistry: rabbit anti-Arid1a (Trilogy, 1:2000, #HPA005456, Sigma)[36], rabbit anti-Pten (Citrate buffer, 1:200, #9559, Cell Signaling Technologies)[37], rabbit anti-phospho Smad3 at S423 and S425 (Trilogy, 1:250, #ab52903, Abcam)[38], and rabbit anti-phospho Akt Ser473 (Trilogy, 1:100, #4060, Cell Signaling Technologies).

**Cell migration and invasion assays.** Cell migration was measured using an in vitro wound-healing assay on 6-well plates. After reaching confluence, cells were serum starved for 24 h prior to the experiment. Cell monolayers were then scratched with a pipette tip to generate a wound. The media and dislodged cells were aspirated, and replaced with fresh serum-free growth medium with or without the addition of 10 ng/mL TGF-β1. Cells were incubated at 37 °C and were allowed to migrate. CytoSMART Lux 10x System (Lonza) was used to record time-lapse images at 15 min intervals from time 0 to 8 h. The width of wounded cell monolayers was measured using ImageJ software, and the rates of migration were calculated.

Cell invasion was determined using the xCELLigence system (ACEA Biosciences). For invasion studies, CIM-plate 16 were pre-coated using growth factor reduced Matrigel (800 μg/mL), and medium containing TGF-β1 (10 ng/mL) was added to the lower chambers, and $ARID1A^{WT}$ or $ARID1A^{KO}$ cells (40,000 cells/well in 1% FBS containing growth medium) were seeded in the upper inserts. Cells were serum starved for 24 h prior to the experiment. In the ARID1A re-expression experiment, ectopic expression of ARID1A in $ARID1A^{KO}$ cells was achieved by transfection of the pcDNA6-V5/HisB plasmid containing wild-type full-length ARID1A[24]. The xCELLigence system detects changes in impedance over time as the cells migrate to the electrode arrays on the bottom side of the upper chamber. Real-time cell invasion was monitored every 10 min for 60 h. Each sample was assayed in three or more replicates. An index of invasion was calculated using RTCA Software Package 1.2.

Chemotaxis assay was performed by seeding $ARID1A^{WT}$ or $ARID1A^{KO}$ cells at $3 \times 10^5$ cells/100 μl onto chemotaxis μ-slides coated with collagen IV. The slide chamber was then flushed with serum-free medium, and the reservoirs on one side were filled with serum-containing medium and the other side with serum-free medium containing TGF-β1 at 10 ng/mL. Live-cell imaging was performed using a Nikon Eclipse TE300 microscope. Data were captured on a Hamamatsu ORCA-ER camera using SimplePCI automated image capture software. Cells were imaged at 10 min intervals at 10× magnification for up to 24 h. Cell migration was manually tracked using CellTracker software and analyzed using ImageJ Chemotaxis and Migration Tool plug-in. A total of 38 cells that did not undergo division during the experiment were tracked. Trajectory plots and chemotactic responses were measured.

**ChIP-seq data processing.** For each sample, single-end ChIP-seq reads were aligned to hg19 by Bowtie v0.12.7[39], and uniquely mapped reads were selected for analysis, using the parameter -m 1. Aligned reads were stored in BAM format using SAMtools v0.1.19[40]. ChIP-seq peaks were called by MACS2 v2.1.0[41] using matched sample and input BAM files. For each transcription factor (ARID1A, BRG1, and RNA Pol II), narrow peaks were called by setting the q-value at 0.01. ChIP-seq peaks were further compared and annotated using HOMER v4.7.2[42]. ARID1A consensus peaks and ARID1A/BRG1 consensus peaks were identified by the tool mergePeaks considering peaks separated by less than 100 bp as common peaks. We annotated each peak using the annotatePeaks.pl function of HOMER. A histogram of read intensities in a 6 kb window around the peak center (option -size 3000 -bin 50) was generated. For each pair of $ARID1A^{WT}$ and $ARID1A^{KO}$ ChIP-seq experiments, we drew their read intensity histograms based on both ARID1A consensus peaks and ARID1A/BRG1 consensus peaks. ChIP-seq read coverage was quantified in bedGraph format by MACS2 with -B option and further converted to bigwig files by deepTools v2.4.3[43]. PCA analysis was performed using the plotPCA function. The first 3 principal components were extracted to plot the figure in three dimensions.

The gene region information for hg19 was extracted from the RefSeq annotation file, which was downloaded from the UCSC genome browser[44], and included 28,830 genes. The ChIP-seq read intensity plots at gene regions were generated by deepTools. In detail, first, ChIP-seq read coverage was quantified using bamCoverage with options --normalizeTo1x 2451960000 --ignoreDuplicates --extendReads 200. Read intensities at gene regions from 3 kb before the transcription start site (TSS) until 3 kb after the transcription end site (TES) were

extracted from the read coverage using the scale-region command of computeMatrix. Here, we scaled the length of gene regions to 5 kb. Finally, read intensity curves were plotted by plotProfile.

**Quantification and statistical analysis**. All individual experiments were performed in triplicate. Data were presented as the means ± standard error of the mean (SEM). Statistical significance to detect difference between two groups was evaluated using paired two-tailed Student $t$-test unless otherwise stated. One-way analysis of variance (ANOVA) with Bonferroni's multiple comparison post-test was performed for comparisons with multiple time points. Difference were considered significant for $p < 0.05$ (*), $p < 0.01$ (**), and $p < 0.001$ (***). All statistical analyses were determined by GraphPad Prism software.

**Reporting summary**. Further information on research design is available in the Nature Research Reporting Summary linked to this article.

## Data availability

The ChIP-seq, ATAC-seq, and RNA-seq have been deposited in the NCBI GEO database under accession code GSE106665. All TCGA that support the findings of this study are available in/from the cBioPortal for Cancer Genomics web portal site, [https://www.cbioportal.org/]. All other data is available in the Article, Supplementary Information or available from the authors upon request.

## Code availability

All codes used in this study are available from open source.

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

## Acknowledgements

This work was supported in part by the NIH/NCI grants P50CA228991, P30CA006973, R21CA165807, R01CA215483, R01CA129080 (I.-M.S.), ACS RSG-18-028-01 (M.I.V.), by the Department of Defense (DoD) grants W81XWH-11-2-0230/OC100517 (I.-M.S. and T.-L.W.), by the Ovarian Cancer Research Alliance, the Honorable Tina Brozman Foundation, the Endometriosis Foundation of America, and the Gray Foundation (I.-M. S. and T.-L.W.), and by the Richard W. TeLinde Endowment Fund from the Department of Gynecology and Obstetrics, Johns Hopkins University (I.-M.S.), The authors thank Asli Bahadirli-Talbott for assistance with immunohistochemistry staining.

## Author contributions

Y.S.R., T.-L.W., and I.-M.S. conceived and designed the study. Y.S.R., W.S., Y.Y., Z.-C.Y., T.M., M.-H.L., V.S., R.A., and M.I.V. acquired data. Y.S.R., X.S., X.C., J.X., T.-L.W., and I.-M.S. performed bioinformatics analysis. Y.S.R., W.S., Y.Y., M.I.V., S.S.M., D.W., R.D., T.-L.W., and I.-M.S. analyzed and interpreted data. R.D. provided critical reagents. G.S. copyedited and proofread the paper. Y.S.R., Y.Y., M.I.V., T.-L.W., and I.-M.S. wrote the paper.

## Competing interests

The authors declare no competing interests.
