## [Peer Review File · Nature Communications]

Reviewers' comments:

Reviewer #1 (Remarks to the Author): Expertise in endometrial cancer models

In Rahmanto et al, the authors tackle through genetic means the mechanisms by which the chromatin remodeller ARID1A serves as a tumor suppressor in the endometrium. They generated a novel mouse model of endometrial carcinogenesis (with striking lethality and short latency) based on Arid1a conditional inactivation through an inducible Pax8-Cre driver.

The study is highly-relevant and the model is physiologic, making the study and model of great importance in clarifying the role of ARID1A as a major tumor suppressor in endometrial cancer. Endometrial cancer is a major cancer which has not received sufficient attention from the research community, and ARID1A is more frequently mutated in this type of cancer than any other (which perhaps the authors ought to point out in their introduction), adding further significance to the study. Also, the biological functions of ARID1A in endometrial biology have been poorly elucidated so far.

Transcriptomic profiling led to the identification of TFGb signalling as an important and novel mediator of the ARID1A knockout phenotype, validated by functional studies and CHIP. The authors further complement these approaches with studies of human cell lines they have engineered to be isogenic for ARID1A. The overall claims are reasonably well-substantiated and statistical analysis appears generally adequate. However, the authors should address the following issues prior to publication:

- 1) The IHC showing efficient KO appears very good but should be shown at much higher magnification as it is difficult to interpret at such low magnification (the Pten inset is useful but not sufficient).
- 2) A PTEN alone phenotype has been amply described in the past, but the ARID1A alone phenotype is not adequately described in this study. This is important because it might lead to insights as to ARID1A normal functions in endometrium. The authors should at least comment on the endometrial histology (is it entirely normal? any evidence of intraepithelial neoplasia? any evidence of increased proliferation or apoptosis?).
- 3) Can the authors show PAX8 IHC on the mouse uteri? Is PAX8 indeed expressed only in epithelium as expected? The authors can also presumably show reporter mouse data (e.g. with fluorescent protein or b-gal reporter) or at least refer to published reference(s).
- 4) Re the p-Akt IHC, I am skeptical that the apparent delay in the Pten mice is relevant. Could the differences be due to estrus cycling? Were the mice synchronized/dated? If not, I might suggest showing the 2 week data alone as the significance of the 1 week result is uncertain especially as the number of biological replicates is not stated.
- 5) The pSMAD3 westerns are convincing, but this reviewer can't learn anything useful from the extremely low-magnification panels of tissue sections subjected to IHC (4A).
- 6) Is there misregulation of ER/PR in the context of ARID1A? Arguably, since steroid hormone signalling is so important to endometrial cancer initiation and progression, ER/PR expression studies, at least by IHC, should be included as part of any new mouse model. Are ER/PR retained following ARID1A KO?

Diego H. Castrillon

Reviewer #2 (Remarks to the Author): Expertise in ARID1A

The study by Rahmanto et al. presents a new mouse model for ARID1A/PTEN deficient endometrial cancer. They report the generation of Pax8-rTA; TetO-Cre crossed with Arid1af/f, Ptenf/f, or Arid1af/f Ptenf/f conditional knockout mice. They find that the combined deletion of Arid1a and Pten results in endometrioid neoplasia by 1 week of age, developing into early carcinoma by 2 weeks and advanced carcinoma by 10-12 weeks, in contrast Arid1a deletion alone, which results in no abnormal growth, and Pten deletion alone, which gives rise to neoplasia and early carcinoma that does not progress (at least by 20 weeks of age). The authors identify the TGF- β pathway as an affected pathway by RNA-seq and observe increased motility and invasion in an Arid1a KO human endometrial epithelial cell line, hEM3, generated previously by CRISPR engineering that is refractory to TGF- β 1 addition. Finally, they perform ARID1A ChIP-seq and ATAC-seq on human cell lines to identify ARID1A target genes, including several members of the TGF- β signaling pathway.

While the mouse model presented is an advance, the manuscript is underdeveloped in many respects regarding experimental comparisons and connection to human disease.

Figure 1: It would be good to show an experimental analysis of the TCGA human endometrial cancer samples to justify the model. How many patients have ARID1A mutation, ARID1A haploinsufficiency, what is the overlap with PTEN mutations/LOH, etc. Do ARID1A and PTEN mutations significantly co-occur?

Figure 2: While experimental cohorts analyzed in Figure 2a are quite compelling, the authors should extend this data to other analyses, for example tumor weight and survival in iPAD, iPD, iAD, and controls. It is not sufficient to show survival +/- dox in the iPAD model alone.

Figure 3: The RNA-seq analysis could be further developed here and some of the supplementary data represented in figure form. One thing that might help is to do RNA-seq from a normal control and derive differentially expressed genes between the control versus iPAD or iPD, as well as an iPAD versus iPD comparison. With the current visualization, the iPAD and iPD models look quite similar (e.g. by Euclidean distance in 3c and 3e), and yet have quite different kinetics of tumor development and metastasis. How many genes are differentially regulated between iPAD and iPD tumors? It would be helpful to include Venn Diagrams of DEGs between all comparisons, heat map clustering, and a list of the top 10 Hallmark enrichment, GO enrichment terms with p values for the mouse and human cell line data. As is, figure 3D is underwhelming...it is also not clear if this is a representative "TGF- β " gene set, or a one-off that happened to show enrichment.

Also, now that RNA-seq data is available for primary endometrial cancers through the TCGA, there is no need to rely on CRISPR-derived cell lines. The authors should use publically available data and perform analyses against their GEM model to see how representative the model is. It would be necessary to provide a complete work-up here of the mouse to human primary tumor comparison: GSEA, GO pathway terms, leading edge genes identified, etc.

Figure 4. a. Is there a way to quantitate this? I'm not sure what I'm looking at (higher magnification would help) and neither ARID1A nor pSMAD look very different between +/- dox.

a/b. Again, is there TCGA RPPA data available for this? That would arguably be a better comparison, with many experimental data points.

h. What is the significance of TGF β 1-dependent inhibition in the Arid1a WT line? and comparing KO to KO+TGF β 1? This effect seems incredibly small.

Figure 5. The ChIP-data and ATAC-seq data look quite well done. However, the analysis of these data could be expanded significantly. Previous reports have shown that ARID1A can both positively regulate enhancers to promote gene expression (Mathur et al Nat Genetics 2017, Kelso Nat Comm 2017) and also repress genes through recruitment of HDACs primarily at promoters (Kim Cell

Reports 2016). To define the mechanism of ARID1A action, the authors should analyze where in the genome ARID1A is most commonly bound (promoters, enhancers, etc), as well as TF motifs underlying ARID1A binding sites. They should perform a comparative analysis of ARID1A loss and ATAC-seq loss (how many sites, which TF motifs, etc). Also, are there any ATAC-seq gains? How do ATAC-seq gains compare with ARID1A binding in the WT? How do ATAC loss and gains relate to changes in gene expression?

It also seems quite doable to perform ATAC-seq from the mouse iPAD and iPD tumors and identify differential changes in accessibility and relate this to changes in gene expression.

Reviewer #3 (Remarks to the Author): expertise in endometrial cancer

Mutational inactivation of the ARID1A tumor suppressor gene and/or loss of expression of the encoded protein are frequent in human endometrioid endometrial cancers; their frequency increases with increasing tumor grade. The aim of the current study was to identify the mechanism(s) by which Arid1a deletion contributes to endometrial tumorigenesis, using genetically engineered mouse models of endometrial cancer and isogenic (ARID1AWT and ARID1AKO) human endometrial cell lines, combined with functional studies of selected (TGF-B) networks dysregulated in these models. The major findings reported are:

- 1) Arida1 deletion and Pten deletion have synergistic effects in endometrial tumorigenesis; Arida1 deletion alone was insufficient to induce tumorigenesis.
- 2) Gene networks that were correlated with Arid1a deletion in murine tumors and with ARID1A knockout in human endometrial (nontumorigenic) cells included TGFB1 and TP53 networks.
- 3) Based on functional studies in human isogenic ARID1AWT and ARID1AKO cells, the study concludes that ARID1AKO cells were more invasive than ARID1AWT cells and invasiveness was suppressed by exogenous TGF-B1.
- 4) ChIP seq experiments led to the identification of 594 potential ARID1A target genes, which included genes within the TGF-B1 pathway.

The authors conclude that ARID1A inactivation releases the brake on cell migration and invasion imposed by TGF-b signaling, and that the ARID1A/TGF-b axis is important in promoting tumor cell infiltration into the myometrial wall.

The topic is important, but there are some concerns with certain elements of the data interpretation and conclusions.

Specific comments (in chronological order)

1. In the title, "ARID1A" should be changed to the mouse gene nomenclature (Arid1a). Same applies for all mouse genes throughout the manuscript and figures.
2. The abstract states that transcriptomic analysis on endometrial cancers and precursors derived from the mouse models showed a close resemblance to human uterine endometrioid endometrial cancer. However, the mouse tumor transcriptomes were compared to the TCGA uterine cancer dataset, which is composed of both uterine endometrioid and uterine serous tumors. How can the authors be sure that the mouse tumor transcriptomes resemblance was specific to the uterine endometrioid tumor transcriptomes and not to the serous or the combined endometrioid/serous tumor transcriptomes?
3. It is unclear when the isogenic human cell lines were generated: Lines 31-32 of the abstract infers that the ARID1AWT and ARID1AKO isogenic cell lines were established within the current study in order to verify findings from the murine models, whereas the methods section (lines 591-592) states that these lines "have been reported previously 21,33". Lines 189-191 cites a method

(ref 24), but does not explicitly state that these lines were established in previously published work (Lines 189-191: "These pairs of ARID1AWT and ARID1AKO isogenic cell lines were established from human endometrial epithelium using a CRISPR/Cas9 genome editing method previously described by our lab24.")

4. The number of new cases and deaths cited for EC are different in the introduction (lines 57-59) and discussion (lines 308-310).

5. The intended use of the term "endometrioid cancer" is unclear throughout the manuscript. Is the intended meaning "endometrial cancer" or "endometrioid endometrial cancer"?

6. The introduction should clarify that there are multiple histotypes of EC, and that PTEN and ARID1A mutations are most frequent in the endometrioid subtype.

7. The introduction notes "previous studies reported significant co-occurrence of ARID1A mutation and PTEN (or PIK3CA) mutation in endometrial tumors, suggesting co-operation between these two gene pathways (Ref 7)." Does this statement remain true after correcting for histological subtype? In other words, does the co-occurrence of ARID1A mutation with PTEN/PI3K mutation in The Cancer Genome Atlas data (Ref 7), which consists of serous and endometrioid EC, simply reflect the fact that these aberrations are prominent in endometrioid EC but rare in serous EC? A query of the TCGA data on the endometrioid histotype (n=200) indicates that the pairwise co-occurrence of ARID1A-PTEN-PIK3CA mutations is not statistically significant (as shown below):

TCGA Endometrioid EC mutation frequency and pattern

TCGA Endometrioid EC mutual exclusivity/co-occurrence test

8. Line 130: Add values (% , n/n) to support the contention that "almost all" iPAD mice develop endometrial carcinoma.

9. Lines 132-133: A description of the morphological and clinical features used to infer similarity of mouse tumors to advanced endometrioid endometrial cancers is needed.

10. Line 137: Remove the approximate (~10%) symbol, since the result of the calculation (3/30) is precise.

11. Lines 140-141: A p-value is needed to support the statement that doxycycline-treated iPAD mice had significantly shorter survival time than untreated iPAD mice.

12. Lines 147-149: It is unclear how the data in Supplementary Figure 2A support "step-wise" tumor progression; clarification is needed regarding the definition of "step-wise" and the rationale for inferring this specific descriptor from the data presented in Supplementary Figure 2A.

13. Lines 177-179: What was the rationale for selecting the 7 TCGA tumor types for comparison to the mouse tumor transcriptomes. i.e. why were these 7 tumor types chosen from the 33 tumor types in TCGA?

14. Lines 177-182: The text states that iPAD and iPD tumors were most similar to TCGA human uterine endometrioid carcinoma. However, Figures 7B and 7C refer to the TCGA human endometrial carcinoma cohort which consists of both endometrioid and serous carcinomas, two

histotypes that are molecularly and clinically distinct. Please clarify whether the entire TCGA uterine cancer cohort (endometrioid and serous) was analyzed, or whether the analysis was restricted to the endometrioid tumors within this cohort.

15. Line 182-184: Clarification is needed as to whether Gene Set Enrichment Analysis (GSEA) was performed using all genes in Supplementary Table 1 as input, or just a subset of those genes. If it was the latter, a column should be added to the table to indicate the input genes, and the rationale for selecting those genes should be provided in the methods and text.

16. Lines 184-186: The text states an FDR of <0.001 (FIG. 3D), whereas the figure itself shows an FDR of "0.000".

17. Line 187: Since PTEN mutation is frequent in normal endometrium, and Arid1a deletion and Pten deletion were synergistic in the mouse models, please provide the PTEN mutation status and PTEN protein expression status of the isogenic human endometrial epithelial cell lines.

18. Supplementary Table S4 indicates that expression analysis was performed for isogenic lines under normal growth conditions and under serum starved conditions. The text and methods should specify whether the data in Figure 3A and in Supplementary Table 4 were generated under normal growth conditions or serum starvation. The serum starvation conditions should be provided in the Methods.

19. Lines 200-201 states "The observation that TGF- β signaling was a component of an ARID1A-regulated network is a novel finding." This is not a novel finding. As the authors later (line 296-298) note, THBS1, a gene within the TGF- β signaling pathway, is a well-known ARID1A target gene. Moreover, a connection between ARID1A and TGF- β already exists in the literature:

(i) Genetics. 2016 Aug;203(4):1601-11.

The Chromatin Remodeling Component Arid1a Is a Suppressor of Spontaneous Mammary Tumors in Mice.

Kartha N, Shen L, Maskin C, Wallace M, Schimenti JC.

"To better understand the mechanism by which restoration/overexpression of Arid1a impairs growth and tumorigenesis, we conducted RNA sequencing (RNA-seq) comparing the transcriptomes of 23116 MT vs. AB-C1 and AB-C2 cells. A total of 554 genes were significantly differentially expressed (DE) between the parental stock and the Arid1a-transduced lines [fragments per kilobase of exon per million fragments mapped (FPKM) > 5 ; $\log_2 > 1$ or < -1 Table S3]. Ingenuity pathway analysis (IPA) of these DE genes revealed that the TRP53 pathway was activated in AB-C1/C2 cells, while the TGF β pathway was repressed (Figure S3)."

(ii) J Pathol. 2016 Jan;238(1):21-30. doi: 10.1002/path.4599. Epub 2015 Sep 15.

Arid1a inactivation in an Apc- and Pten-defective mouse ovarian cancer model enhances epithelial differentiation and prolongs survival.

Zhai Y, Kuick R, Tipton C, Wu R, Sessine M, Wang Z, Baker SJ, Fearon ER, Cho KR.

"We employed oligonucleotide microarrays to compare gene expression in Apc-;Pten- versus Apc-;Pten-;Arid1a- mouse tumours (n = 3 independent tumours for each of the two groups). Using log-transformed data on 45 101 probe sets, we selected those probe sets for which two-sample t-tests gave $p < 0.01$ and the difference in means was at least 1.3-fold, to obtain 3895 differentially expressed probe sets (1935 up and 1960 down in Arid1a-/- tumours). We similarly analysed datasets that were identical except that the sample labels were permuted, and obtained just 61.2 qualifying probe sets on average, so that a simple estimate of the false discovery rate for this selection was 1.6% ($=61.2/3895$). Figure 4 shows a small subset of these selected differences, for which average changes were at least 10-fold. The complete dataset and analysis is publicly available in GEO. We collapsed our selections to distinct genes and performed enrichment testing against 3000 curated gene sets from MSigDB. The five top gene sets for lists of genes up- and down-regulated in tumours with bi-allelic Arid1adeletion are shown in Table S1 (see

supplementary material). Notably, our down-regulated gene list's best enrichment ($p = 6E-15$) was for a list of genes higher in luminal versus mesenchymal breast cell line samples 27, whereas our up-regulated gene list's second best enrichment was to the corresponding list of genes higher in mesenchymal genes from the same study ($p = 1.8E-21$). Similarly, when we assessed Gene Ontology (GO) biological process terms, the third-best term for our down-regulated genes was 'positive regulation of epithelial-mesenchymal transition' (10 of 20 genes, $p = 8E-8$), including Axin2, Bmp2, Ctnnb1, Tgfb2 and Tgfb3, and our top hit was to 'cell adhesion' ($p = 9E-12$), which included Fn1, being down over 5-fold. For our down-regulated genes, 'Wnt signalling pathway' was the second most significant enrichment in KEGG pathways and the ninth most significant in GO. Wnt5a was in these intersections and was decreased by 76-fold in our Arid1a-deficient mouse tumours. Levels of Axin2, an indicator of Wnt pathway activation and transcriptional target of β -catenin, were decreased approximately 2-fold."

20. Lines 207-211: I am not convinced, based on the data presented in Figure 4A, that there is weaker phospho-SMAD3 immunoreactivity in uterine epithelial cells of iAD mice with induced ARID1A deletion (iAD+doxycycline) than in mice without induced knockout (iAD-doxycycline). One iAD+doxycycline panel looks to have reduced phospho-SMAD3 staining but the other two iAD + doxycycline panels look to have the same degree of immunoreactivity as the three uninduced (iAD minus doxycycline) panel.

21. Lines 239-242 state that transient re-expression of ectopic ARID1A in ARID1AKO cells restored their sensitivity to TGF-B1, and that upon exposure to TGF-B1 there was a clear reduction in invasive capacity of these cells (Supplementary Fig. 4A). The legend to Supplementary Figure 4A states "re-expression of ARID1A in ARID1AKO decreased the ability of cells to invade through the Matrigel layer (black line, right panel) in comparison to the parental ARID1AKO cells (red) and ARID1AKO cells transfected with pcDNA6 empty vector (gray)."

No explanation is offered as to why the vector-only control (pcDNA6 empty vector) (gray) decreased the ability of cells to invade through the Matrigel layer compared to the parental cells. Are the migration differences between the vector control and the ARID1A expression construct statistically significant? Is the difference in migration between cells transfected with empty vector compared to those expressing vector with ARID1A insert really due to the biological activity of ARID1A, or could it instead be related to the increased burden of harboring a large plasmid/insert, since the empty vector alone caused an effect compared to the parental cells. In other words, does transfecting the parental cells with increasingly large plasmids reduce their invasive capacity? Given these concerns, the statement (in the abstract) that ARID1A-deficiency leads to loss of TGF-b tumor suppressive function seems like an over-interpretation of the data.

22. The discussion should clearly acknowledge that the observed synergy between ARID1A deletion and PTEN deletion in endometrial tumorigenesis may not be completely attributable to altered TGF-b signaling, since many pathways and networks other than TGF-B were transcriptionally dysregulated in iPAD tumors compared to the iPD tumors.

23. Line 347-349: What specific piece of data supports the conclusion that the ARID1A/TGF-b axis is important in promoting tumor cell infiltration into the myometrial wall?

24. Figure 1A: The cartoons of protein expression (right hand side) are not very meaningful; it would be better to show the actual IHC results in this panel (currently provided in Supplementary Figure 1). It would also be helpful to include an "n" value in the figure, to note the number ("n") of animals in each group.

25. Figure 2D: Please add a notation or data to clarify the status of iPAD control mice at weeks 16 and 20. In other words, please note whether or not uterine sampling was performed at these time points for iPAD controls.

26. Figure 2D: It is unclear why the Y-axis extends to 125%, since % survival cannot exceed 100%.

Reviewer #4 (Remarks to the Author): expertise in TGFbeta signalling

This is an interesting and well performed study. The authors generate an exciting new mouse model of endometrial cancer combining Arid1a deletion with PTEN loss. Transcriptional interrogation of this model along with human endometrial cell lines with deletion of Arid1a suggests intimate relationships between Arid1a and TGF β signaling whereby the authors postulate that a functionally relevant consequence of Arid1a loss is modulation of TGF β signaling. This is exciting and novel.

I have several points which I would like the authors to consider.

1) Discussion and title.

The authors claim that TGF β mediates the in-vivo tumour suppressor activity of Arid1A but they did not actually show this in this paper. To show this definitively would require experimental maintenance of TGF β signaling in the absence of Arid1a (for example by overexpression of TGFBR2 to WT levels in the Arid1a ko model) and a block to tumour promotion. Similarly, TGF β signaling inhibitors should result in a similar promotion of tumourigenesis in the PTEN ko model and this should be epistatic to Arid1a loss. These experiments would be very time consuming in-vivo so I would suggest that the authors re-word their title and discussion to reflect the correlative links rather than cause and effect. Some in-vitro experiments suggested below should be informative.

2) The mouse model is based on Arid1a and PTEN loss. It would be good to show the overlap and frequency of these events as an oncoprint of published NGS of human endometrial tumours.

3) In figures 1 and 2 the authors describe disseminated tumour nodules how do they know they are not multifocal separate initiation events? It would be good to stain these potential METS for PAX8 also. It would also be useful to provide Kaplan Meier analysis and quantification and location of disseminated disease in their model.

4) In Figure 3 the GSEA analysis and IPA analysis indicates that TGF β pathway is modulated in both the mouse model and the human cells lines. The authors should provide the GSEA analysis figure of the human data alongside that of the mouse (shown in Fig 3D). This may I highlight that actually in the mouse the TGF β signaling pathway is enriched but in the human cell lines loss of Arid1a results in a down regulation of TGF β pathway signaling. The authors should comment on this. It would be interesting to determine if loss of PTEN in the human cell lines would change the TGF β pathway output to more closely resemble the mouse data.

5) A common event between the mouse and human RNAseq data is that TGFBR2 is downregulated. This should be investigate in more detail. Q-PCR validation in both sets of samples along with western blotting would be good. It would also be good to interrogate human RNAseq data from patient material to see if this also occurs during tumour progression and if this correlates with Arid1a and/or PTEN loss. To test if loss of TGFBR2 is responsible for a downregulation of exogenous TGF β response the authors should restore TGFBR2 levels to wild type levels in the human cell lines and measure the effect in PO4-Smad readouts.

Technical concerns

1) The authors should provide validation of their antibodies in IHC conditions for PO4-Smad3, Arid1a and PTEN by embedding cells +/- the antigens (knockout or loss for Arid1a, PTEN and +/- TGFBR inhibitor treated cells for PO4-Smad3) in FFPE conditions.

2) The authors should provide more detail for how the murine tumours were classified into

different grades with clear images of examples provided in the supplementary data.

3) In Figure 3 the authors state that there is a reduction in intrinsic PO-Smad3 levels in the arid1a deficient mouse tissues. The images are unclear, blow up images to show nuclear PO4-Smad3 would be helpful and a histoscore quantification of reduction should be performed complete with statistical analysis. The baseline intrinsic signalling in the knockout human cells is not statistically reduced and the statistical significance of the reduction in exogenous ligand stimulation should be analysed.

4) In Supp Figure 4a the authors claim that restoration of ARID1A to ko cells restores TGF β 1 sensitivity. This should be measured by quantitative measurement of PO4-Smad3 in western blot analysis in response to exogenous ligand along with measurement of TGFBR2 levels. The data of the ARID1A ko cells +/- TGF β 1 stimulation, +/- Arid1A restoration should be shown in the invasion assays and assessed statistically.

5) In Figure 4G the authors state that extracellular TGF β 1 induced a significant inhibition of forward migration in wt but not KO cells. This was not statistically assessed in the figure and should be.

Minor points/errors

1) Define UT in Figure 1B legend.

2) Refer to the blue star in Figure 2b

3) The PCA analysis is not shown in Figure 5 and mislabelling/reference in the text to Figure 5 has occurred.

Reviewer #1 (Remarks to the Author): Expertise in endometrial cancer models

R1, Q1: 1) The IHC showing efficient KO appears very good but should be shown at much higher magnification as it is difficult to interpret at such low magnification (the Pten inset is useful but not sufficient).

Response: We added an additional high magnification inset in Supplementary Figure 1A-C.

R1, Q2: 2) A PTEN alone phenotype has been amply described in the past, but the ARID1A alone phenotype is not adequately described in this study. This is important because it might lead to insights as to ARID1A normal functions in endometrium. The authors should at least comment on the endometrial histology (is it entirely normal? any evidence of intraepithelial neoplasia? any evidence of increased proliferation or apoptosis?).

Response: We have described the phenotype of *Arid1a* loss in endometrium (*iAD* mouse) in which we observed normal-appearing endometrium up to 24 weeks. Please see lines 115-117 and 121-123, and Figure 2A for details. In short, there is no evidence of intraepithelial lesions or other detectable abnormalities observed by experienced gynaecologic pathologists.

R1, Q3: 3) Can the authors show PAX8 IHC on the mouse uteri? Is PAX8 indeed expressed only in epithelium as expected? The authors can also presumably show reporter mouse data (e.g. with fluorescent protein or b-gal reporter) or at least refer to published reference(s).

Response: The Pax8 immunoreactivity in normal mouse uterus is shown below and it is exclusively expressed in all epithelial cells but not in stromal, endothelial or inflammatory cells within the uteri. More importantly, we also performed Pax8 IHC in mouse endometrial tumor lesions and found Pax8 staining, like their normal counterparts, is exclusive detected in tumor cells. Since Pax8 expression pattern is well known in normal uteri, we used Figure 2B-C to shows Pax8 immunoreactivity in uterine tumor epithelial cells. We believe that this would be more informative.

R1, Q4: 4) Re the p-Akt IHC, I am skeptical that the apparent delay in the Pten mice is relevant. Could the differences be due to estrus cycling? Were the mice synchronized/dated? If not, I might suggest showing the 2 week data alone as the significance of the 1 week result is uncertain especially as the number of biological replicates is not stated.

Response: We agree to show only the 2-week data alone and revised the manuscript accordingly.

R1, Q5: 5) The pSMAD3 Westerns are convincing, but this reviewer can't learn anything useful from the extremely low-magnification panels of tissue sections subjected to IHC (4A).

Response: We have added an additional higher magnification inset in Supplementary Figure 4B.

R1, Q6: 6) Is there misregulation of ER/PR in the context of ARID1A? Arguably, since steroid hormone signalling is so important to endometrial cancer initiation and progression, ER/PR

expression studies, at least by IHC, should be included as part of any new mouse model. Are ER/PR retained following ARID1A KO?

Response: We have included an additional description (lines 123-125) of ER/PR IHC performed on ARID1A KO mice (*iAD*) in the Revised Result section. We did not observe significant changes in ER/PR expression patterns in the endometrium of all *iAD* mice examined as compared to the wildtype.

Reviewer #2 (Remarks to the Author): Expertise in ARID1A

R2, Q1: Figure 1: It would be good to show an experimental analysis of the TCGA human endometrial cancer samples to justify the model. How many patients have ARID1A mutation, ARID1A haploinsufficiency, what is the overlap with PTEN mutations/LOH, etc. Do ARID1A and PTEN mutations significantly co-occur?

Response: We now added data on TCGA UCEC analysis which can be found in Figure 1B. The analysis indicates a significant co-occurrence between *ARID1A* and *PTEN* mutations.

R2, Q2: Figure 2: While experimental cohorts analyzed in Figure 2a are quite compelling, the authors should extend this data to other analyses, for example tumor weight and survival in *iPAD*, *iPD*, *iAD*, and controls. It is not sufficient to show survival +/- dox in the *iPAD* model alone.

Response: We agree in principal and have indeed included *iPAD* +/- dox survival data (Figure 2D). We did not include survival data for *iPD* and *iAD* because both mice were alive at all time points tested (Figure 2A).

With regards of tumor weight, in *iPAD* mouse tumor develop in the uteri and invade/spread into peritoneal cavity (Figure 1B). Thus, it would be highly challenging to accurately measure tumor weight in disseminated tumors. Instead, we provide a representative image of the tumor at necropsy.

R2, Q3: Figure 3: The RNA-seq analysis could be further developed here and some of the supplementary data represented in figure form. One thing that might help is to do RNA-seq from a normal control and derive differentially expressed genes between the control versus *iPAD* or *iPD*, as well as an *iPAD* versus *iPD* comparison. With the current visualization, the *iPAD* and *iPD* models look quite similar (e.g. by Euclidean distance in 3c and 3e), and yet have quite different kinetics of tumor development and metastasis. How many genes are differentially regulated between *iPAD* and *iPD* tumors? It would be helpful to include Venn Diagrams of DEGs between all comparisons, heat map clustering, and a list of the top 10 Hallmark enrichment, GO enrichment terms with p values for the mouse and human cell line data. As is, figure 3D is underwhelming...it is also not clear if this is a representative "TGF-b" gene set, or a one-off that happened to show enrichment.

Response: The Euclidian distance is a relative measurement between *iPAD*, *iPD*, and other human tumor (TCGA data). Therefore, *iPAD* and *iPD* models appear to be relatively similar to each another because we included a wide variety of tumor types. We identified 1,156 DEGs between *iPAD* and *iPD* tumors (please see lines 163-166 and Supplementary Table 1). Moreover, we performed "Ingenuity IPA top upstream network regulators analysis" on our mouse and human DEGs to identify TGFB1 were shared between the mouse and human networks (please see lines 166-169, 193-197, and Supplementary Table 2 and 4) and have included the p-value.

R2, Q4: Also, now that RNA-seq data is available for primary endometrial cancers through the TCGA, there is no need to rely on CRISPR-derived cell lines. The authors should use publically available data and perform analyses against their GEM model to see how representative the model is. It would be necessary to provide a complete work-up here of the mouse to human primary tumor comparison: GSEA, GO pathway terms, leading edge genes identified, etc.

Response: To determine the role of *ARID1A* mutations in endometrial cancer carcinogenesis, we used *ARID1A* CRISPR knockout on primary human endometrial epithelial cells to mimic *ARID1A* mutations which lead to loss of *ARID1A* protein. We hope that this provides a "clean" model (only *ARID1A* is deleted) which complements to the TCGA tumors which harbor a variety of genetic alterations in addition to *ARID1A* mutation. To answer the 2nd question, we have used publically

available data and performed Euclidian distance analysis on our mouse RNA-seq data and demonstrated that our GEM model is most similar to UCEC dataset with respect to 7 TCGA carcinomas used in the analysis (please see Figure 3B and 3C).

R2, Q5: Figure 4. a. Is there a way to quantitate this? I'm not sure what I'm looking at (higher magnification would help) and neither ARID1A nor pSMAD look very different between +/- dox.

Response: In addition to original photomicrographs to show the pattern in histopathology and to support the absence of lesion, we have added images with higher magnification in Figure 4A which clearly shows a reduction of immunoreactivity.

R2, Q6: a/b. Again, is there TCGA RPPA data available for this? That would arguably be a better comparison, with many experimental data points.

Response: Our IHC and Western blot (Figure 4A-B) assessed changes in the phosphorylation level of SMAD3 protein. Thus, the use of TCGA RPPA data appear not informative for phosphorylation levels, as TCGA RPPA only contains total protein data.

R2, Q7: What is the significance of TGFb1-dependent inhibition in the Arid1a WT line? and comparing KO to KO+TGFb1? This effect seems incredibly small.

Response: The single cell tracking experiment was designed to compare differences between *ARID1A^{WT}* and *ARID1A^{KO}* cells in the presence and separately in the absence of TGF-β1. We detected a trend that *ARID1A^{WT}* cells move away from a TGF-β1 gradient. Importantly, using other assays with higher cell density, we observed a significant difference in the *ARID1A^{WT}* cells showing reduced wound closure rate and invasion in the presence of TGF-β1. We have revised our statement as we observed a modest but statistically significant inhibition of forward migration in *ARID1A^{WT}* cells as compared to *ARID1A^{KO}* cells in the presence of TGF-β1 (lines 251-254).

R2, Q8: Figure 5. The ChIP-data and ATAC-seq data look quite well done. However, the analysis of these data could be expanded significantly. Previous reports have shown that ARID1A can both positively regulate enhancers to promote gene expression (Mathur et al Nat Genetics 2017, Kelso Nat Comm 2017) and also repress genes through recruitment of HDACs primarily at promoters (Kim Cell Reports 2016). To define the mechanism of ARID1A action, the authors should analyze where in the genome ARID1A is most commonly bound (promoters, enhancers, etc), as well as TF motifs underlying ARID1A binding sites. They should perform a comparative analysis of ARID1A loss and ATAC-seq loss (how many sites, which TF motifs, etc). Also, are there any ATAC-seq gains? How do ATAC-seq gains compare with ARID1A binding in the WT? How do ATAC loss and gains relate to changes in gene expression?

Response: As suggested, we have added more analyses on ARID1A binding location and TF motifs (Supplementary Figure 5C-D; lines 273-275; lines 277-280). Comparing ATAC-seq signals before and after *ARID1A* knockout, we identified 15,192 (38.6%) of ATAC-seq peak losses, of which 8,979 (~60%) peaks were associated with ARID1A-specific binding events (please see figure below). We also found that 496 ATAC-seq signals were gained in response to *ARID1A* loss, of which 33 peaks were associated with ARID1A-specific peaks (please see figure below). Moreover, ATAC loss was more significantly associated with differential genes after *ARID1A* knockout than the ATAC gain.

R2, Q9: It also seems quite doable to perform ATAC-seq from the mouse iPAD and iPD tumors and identify differential changes in accessibility and relate this to changes in gene expression.

Response: We agree that this is a good idea but also recognize that this would require extensive additional studies and analyses. We plan to perform such experiments with a comprehensive design and deeper analysis in an independent study in the near future.

Reviewer #3 (Remarks to the Author): expertise in endometrial cancer

R3, Q1: 1. In the title, “ARID1A” should be changed to the mouse gene nomenclature (*Arid1a*). Same applies for all mouse genes throughout the manuscript and figures.

Response: We have corrected the nomenclature throughout the manuscript. “*Arid1a*” is used to refer to mouse gene, while “*ARID1A*” is used in citing the human counterpart. In the case that we refer to both mouse and human data or the universal functions, we will use the “*ARID1A*”.

R3, Q2: 2. The abstract states that transcriptomic analysis on endometrial cancers and precursors derived from the mouse models showed a close resemblance to human uterine endometrioid endometrial cancer. However, the mouse tumor transcriptomes were compared to the TCGA uterine cancer dataset, which is composed of both uterine endometrioid and uterine serous tumors. How can the authors be sure that the mouse tumor transcriptomes resemblance was specific to the uterine endometrioid tumor transcriptomes and not to the serous or the combined endometrioid/serous tumor transcriptomes?

Response: We have now separated the tumors corresponding to endometrioid and serous subtypes of TCGA UCEC PanCancer Atlas using the Euclidian distance analyses (Figure 3B-C). This new analysis demonstrates that transcriptomes from both mouse tumours (*iPD* and *iPAD*) have the closest resemblance to uterine endometrioid followed by uterine serous (Figure 3B-C).

R3, Q3: 3. It is unclear when the isogenic human cell lines were generated: Lines 31-32 of the abstract infers that the *ARID1AWT* and *ARID1AKO* isogenic cell lines were established within the current study in order to verify findings from the murine models, whereas the methods section (lines 591-592) states that these lines “have been reported previously 21,33”. Lines 189-191 cites a method (ref 24), but does not explicitly state that these lines were established in previously published work (Lines 189-191: “These pairs of *ARID1AWT* and *ARID1AKO* isogenic cell lines cell lines were established from human endometrial epithelium using a CRISPR/Cas9 genome editing method previously described by our lab24.”)

Response: We apologize for the confusion. We have an established isogenic *ARID1A^{KO}* line which was published previously. In this study, we generated a second isogenic *ARID1A^{KO}* line using the same CRISPR strategy. We have revised the wording in the abstract, corrected the reference, and described the second isogenic line in the Cell Culture section of the Methods.

R3, Q4: 4. The number of new cases and deaths cited for EC are different in the introduction (lines 57-59) and discussion (lines 308-310).

Response: We have revised the statement referencing to the latest publication. We only show this fact in the Introduction as there is no need to reiterate it (lines 55-56).

R3, Q5: 5. The intended use of the term “endometroid cancer” is unclear throughout the manuscript. Is the intended meaning “endometrial cancer” or “endometrioid endometrial cancer”?

Response: In the revised manuscript, we used “endometrioid carcinoma of the uterus” for clarity.

R3, Q6: 6. The introduction should clarify that there are multiple histotypes of EC, and that PTEN and *ARID1A* mutations are most frequent in the endometrioid subtype.

Response: We have added further clarification in the introduction as suggested and it can be found on lines 61-65.

R3, Q7: 7. The introduction notes “previous studies reported significant co-occurrence of ARID1A mutation and PTEN (or PIK3CA) mutation in endometrial tumors, suggesting co-operation between these two gene pathways (Ref 7).” Does this statement remain true after correcting for histological subtype? In other words, does the co-occurrence of ARID1A mutation with PTEN/PI3K mutation in The Cancer Genome Atlas data (Ref 7), which consists of serous and endometrioid EC, simply reflect the fact that these aberrations are prominent in endometrioid EC but rare in serous EC? A query of the TCGA data on the endometrioid histotype (n=200) indicates that the pairwise co-occurrence of ARID1A-PTEN-PIK3CA mutations is not statistically significant (as shown below):

TCGA Endometrioid EC mutation frequency and pattern
TCGA Endometrioid EC mutual exclusivity/co-occurrence test

Response: In this study, we focused on the endometrioid subtype only. In our view, statistical significance is not reached for co-occurrence because *PTEN* mutation is so prevalent and only a few cases of endometrioid carcinoma of the uterus don't have *PTEN* inactivation. The important fact is that many, if not almost all, endometrioid carcinomas have inactivation mutations in both the *ARID1A* and *PTEN* pathways.

Moreover, we have added separate data analyses on endometrioid (n = 399) and serous (n = 109) subtypes of the TCGA UCEC PanCancer Atlas (Figure 1B), which has a bigger sample size. Indeed, the analyses indicates that *ARID1A/PTEN* aberrations are more prominent in endometrioid (54%/81%) than in the serous (9%/14%) subtypes. The reference to the TCGA PanCancer Atlas has been added.

R3, Q8: 8. Line 130: Add values (% , n/n) to support the contention that “almost all” *iPAD* mice develop endometrial carcinoma.

Response: This has been done in the revised manuscript.

R3, Q9: 9. Lines 132-133: A description of the morphological and clinical features used to infer similarity of mouse tumors to advanced endometrioid endometrial cancers is needed.

Response: Morphologically, the *iPAD* mouse tumors exhibit confluent glandular growth pattern characteristic to human uterine endometrioid carcinoma. These tumors are initially confined to the uterine mucosa, then infiltrating into uterine myometrium before penetrating through the uterine wall to peritoneal cavity. Metastatic carcinomas can be detected as well at later stages. Therefore, both morphological features and clinical behaviour of the mouse tumors resemble human counterparts.

We have described these features in the sentence following the one that the reviewer referred to in line 133-138 (in the revised manuscript): “These tumors progressed rapidly to highly invasive carcinoma with morphological and clinical features similar to advanced endometrioid carcinoma. Myometrial and angiolymphatic invasions followed by tumor dissemination to the intraperitoneal cavity, and lymph node metastasis, which are characteristics of advanced endometrioid carcinoma, were observed in tumors from *iPAD* mice 6 weeks after doxycycline injection (Fig. 2A).”

R3, Q10: 10. Line 137: Remove the approximate (~10%) symbol, since the result of the calculation (3/30) is precise.

Response: This has been corrected.

R3, Q11: 11. Lines 140-141: A p-value is needed to support the statement that doxycycline-treated iPAD mice had significantly shorter survival time than untreated iPAD mice.

Response: p-value is now added on line 143 in the revised manuscript.

R3, Q12: 12. Lines 147-149: It is unclear how the data in Supplementary Figure 2A support “step-wise” tumor progression; clarification is needed regarding the definition of “step-wise” and the rationale for inferring this specific descriptor from the data presented in Supplementary Figure 2A.

Response: We agree and have deleted the word “stepwise”. The figure is not intended to show step-wise progression.

R3, Q13: 13. Lines 177-179: What was the rationale for selecting the 7 TCGA tumor types for comparison to the mouse tumor transcriptomes. i.e. why were these 7 tumor types chosen from the 33 tumor types in TCGA?

Response: We selected the 7 carcinomas based on pathological similarity to endometrioid carcinomas studied in this report such as a glandular growth pattern. Thus, sarcoma, lymphoma, and squamous carcinoma are not relevant comparisons as they do not exhibit a glandular growth pattern. In this figure panel, we intended to demonstrate that the mouse and human endometrioid carcinomas are molecularly closer than the selected carcinomas given that both mouse carcinoma and human endometrioid carcinoma of the uterus are morphologically very similar. A clarification sentence has been added in lines 175-177.

R3, Q14: 14. Lines 177-182: The text states that iPAD and iPD tumors were most similar to TCGA human uterine endometrioid carcinoma. However, Figures 7B and 7C refer to the TCGA human endometrial carcinoma cohort which consists of both endometrioid and serous carcinomas, two histotypes that are molecularly and clinically distinct. Please clarify whether the entire TCGA uterine cancer cohort (endometrioid and serous) was analyzed, or whether the analysis was restricted to the endometrioid tumors within this cohort.

Response: We have separated the endometrioid and serous subtypes of the TCGA UCEC PanCancer Atlas in the Euclidian distance analyses (Figure 3B and 3C). This new analysis demonstrates that transcriptomes of both mouse tumour types (*iPD* and *iPAD*) have the closest resemblance to uterine endometrioid followed by uterine serous (Figure 3B and 3C).

R3, Q15: 15. Line 182-184: Clarification is needed as to whether Gene Set Enrichment Analysis (GSEA) was performed using all genes in Supplementary Table 1 as input, or just a subset of those genes. If it was the latter, a column should be added to the table to indicate the input genes, and the rationale for selecting those genes should be provided in the methods and text.

Response: We performed GSEA analysis on TGFB1 target signature against all genes in Supplementary Table 1. We have clarified it in the revised text. Please see lines 180-182.

R3, Q16: 16. Lines 184-186: The text states an FDR of <0.001 (FIG. 3D), whereas the figure itself shows an FDR of “0.000”.

Response: We have modified the text accordingly.

R3, Q17: 17. Line 187: Since PTEN mutation is frequent in normal endometrium, and Arid1a deletion and Pten deletion were synergistic in the mouse models, please provide the PTEN

mutation status and PTEN protein expression status of the isogenic human endometrial epithelial cell lines.

Response: We have examined PTEN protein expression and observed similar levels of expression in the isogenic human endometrial epithelial cell lines. Mutation analysis of *PTEN* indicates there is no mutation acquired in these cell lines. This new information has been reported in the Supplemental Figure 4A.

R3, Q18: 18. Supplementary Table S4 indicates that expression analysis was performed for isogenic lines under normal growth conditions and under serum starved conditions. The text and methods should specify whether the data in Figure 3A and in Supplementary Table 4 were generated under normal growth conditions or serum starvation. The serum starvation conditions should be provided in the Methods.

Response: We have clarified the text (lines 192-193) and stated that the same results were obtained in both normal growth and serum-starved conditions. We identified consistently ARID1A-regulated DEGs in both normal and stress conditions (Supplementary Table 3). Thus, the data presented in Figure 3A and Supplementary Table 4 should not be affected, as the DEGs' direction (upregulation or downregulation) were consistent in the two isogenic clones and under both normal and stress conditions. We have added description of the serum starved condition in the Methods section (lines 733-735).

R3, Q19: 19. Lines 200-201 states "The observation that TGF- β signaling was a component of an ARID1A-regulated network is a novel finding." This is not a novel finding. As the authors later (line 296-298) note, THBS1, a gene within the TGF- β signaling pathway, is a well-known ARID1A target gene. Moreover, a connection between ARID1A and TGF- β already exists in the literature:

Response: We acknowledge that the relationship of TGF- β and ARID1A has been reported in the past. However, our study is unique in the detail and depth with which it interrogates TGFB1 and ARID1A interaction. Accordingly, we have modified the sentence as below (please also see lines 203-206) to reflect that our study is the first one to comprehensively characterize the molecular connection between ARID1A and TGF- β pathway and our data provide strong mechanism-driving support to the previous findings that TGF- β related genes were altered according to ARID1A status.

"Our molecular studies as discussed above illustrated TGF- β signaling as a component of an ARID1A-regulated network. Moreover, previous studies suggest TGF- β signaling as a component of an ARID1A-regulated network^{27, 28}, which prompted us to further interrogate whether ARID1A pathway causally affects TGF- β expression and its related phenotypes."

R3, Q20: 20. Lines 207-211: I am not convinced, based on the data presented in Figure 4A, that there is weaker phospho-SMAD3 immunoreactivity in uterine epithelial cells of iAD mice with induced ARID1A deletion (iAD+doxycycline) than in mice without induced knockout (iAD-doxycycline). One iAD+doxycycline panel looks to have reduced phospho-SMAD3 staining but the other two iAD + doxycycline panels look to have the same degree of immunoreactivity as the three uninduced (iAD minus doxycycline) panel.

Response: We are sorry to present the low magnification originally as the histology at such low magnification is not intended to appreciate the staining intensity. Thus, we have added an additional higher magnification panel in Supplementary Figure 4B to demonstrate weaker pSmad3 staining in *iAD* mice (induced *Arid1a*-deletion). Moreover, Figure 4B and 4C provide further support to IHC data. We keep the original Figure 4A (the low power panel) to show the *Arid1a* knockout efficiency (100% loss in epithelium).

R3, Q21: 21. Lines 239-242 state that transient re-expression of ectopic ARID1A in ARID1AKO cells restored their sensitivity to TGF-B1, and that upon exposure to TGF-B1 there was a clear reduction in invasive capacity of these cells (Supplementary Fig. 4A). The legend to Supplementary Figure 4A states “re-expression of ARID1A in ARID1AKO decreased the ability of cells to invade through the Matrigel layer (black line, right panel) in comparison to the parental ARID1AKO cells (red) and ARID1AKO cells transfected with pcDNA6 empty vector (gray).”

No explanation is offered as to why the vector-only control (pcDNA6 empty vector) (gray) decreased the ability of cells to invade through the Matrigel layer compared to the parental cells. Are the migration differences between the vector control and the ARID1A expression construct statistically significant? Is the difference in migration between cells transfected with empty vector compared to those expressing vector with ARID1A insert really due to the biological activity of ARID1A, or could it instead be related to the increased burden of harboring a large plasmid/insert, since the empty vector alone caused an effect compared to the parental cells. In other words, does transfecting the parental cells with increasingly large plasmids reduce their invasive capacity? Given these concerns, the statement (in the abstract) that ARID1A-deficiency leads to loss of TGF-b tumor suppressive function seems like an over-interpretation of the data.

Response: We have added statistical analysis of the data to Supplementary Figure 4C. Based on statistical analysis, we did not observe significant difference between the vector-only control and the parental cell (n.s.). The Matrigel invasion rate was significantly different ($p < 0.05$) between vector-only control cells and ARID1A re-expression cells in the presence of TGF β 1. The size of the transfected plasmid does not apparently affect the invasive capacity of the cells because our control experiment (please see the figure below) using the vector-only control and ARID1A expression plasmid in the absence of TGF β 1 demonstrated no significant difference (n.s.) between the two conditions, or between these conditions and the parental cells. That means the difference between control and ARID1A expressing cells can be appreciated only in the presence of TGF β 1.

R3, Q22: 22. The discussion should clearly acknowledge that the observed synergy between ARID1A deletion and PTEN deletion in endometrial tumorigenesis may not be completely attributable to altered TGF-b signaling, since many pathways and networks other than TGF-B were transcriptionally dysregulated in iPAD tumors compared to the iPD tumors.

Response: We agree and have added this clarification which can be found on lines 340-346:

“Several functional pathways critical for endometrial tumorigenesis are also enriched in ARID1A-target genes include VEGF, IL-8, Stem Cell Pluripotency, and TGF- β (Supplementary Table 8). Thus, the observed synergy between *ARID1A* and *PTEN* deletion in endometrial tumorigenesis cannot be solely attributed to the alteration in TGF- β signaling. Aberrations in other functional pathways resulting from *ARID1A* inactivating mutations may well account for the observed

aggressive phenotype in the *iPAD* mouse model and human endometrioid carcinomas.”

R3, Q23: 23. Line 347-349: What specific piece of data supports the conclusion that the ARID1A/TGF- β axis is important in promoting tumor cell infiltration into the myometrial wall?

Response: We showed clearly that without *Arid1a* co-deletion, *iPD* mouse tumors (*Pten* deletion only) are confined only to mucosa and do not invade into the myometrium. So, we now revised this sentence (lines 352-354) to “Taken together, these data suggest that ARID1A may affect the TGF- β axis, which in turn contributes to tumor cell infiltration into and through the myometrial wall, a critical step in metastasis.”.

R3, Q24: Figure 1A: The cartoons of protein expression (right hand side) are not very meaningful; it would be better to show the actual IHC results in this panel (currently provided in Supplementary Figure 1). It would also be helpful to include an “n” value in the figure, to note the number (“n”) of animals in each group.

Response: We agree and have deleted this part of our cartoons. We are concerned about the IHC photomicrographs which may be too small to be appreciated, so instead, we use + or – to present the presence or absence of gene and gene product. Figure 1A is a schema intended to introduce and provide clarity to readers on the design of the three murine models generated in this study. The number of mice (“n” value) used for pathological assessment can be referred to in Figure 2A whereby each circle represents an individual uterine sample from a mouse.

R3, Q25: 25. Figure 2D: Please add a notation or data to clarify the status of *iPAD* control mice at weeks 16 and 20. In other words, please note whether or not uterine sampling was performed at these time points for *iPAD* controls.

Response: We have added a clarification on the status of *iPAD* control mice at week 16 and 20 in the legend of Figure 2A: “Pathological assessment was not performed on *iPAD* mice at weeks 16 and 20 due to cancer related morbidity and mortality. All *iPAD* control mice were alive and active at both time points.”

R3, Q26: 26. Figure 2D: It is unclear why the Y-axis extends to 125%, since % survival cannot exceed 100%.

Response: This has been fixed.

Reviewer #4 (Remarks to the Author): expertise in TGFbeta signalling

R4, Q1: 1) Discussion and title.

The authors claim that TGF β mediates the in-vivo tumour suppressor activity of Arid1A but they did not actually show this in this paper. To show this definitively would require experimental maintenance of TGF β signaling in the absence of Arid1a (for example by overexpression of TGFBR2 to WT levels in the Arid1a ko model) and a block to tumour promotion. Similarly, TGF β signaling inhibitors should result in a similar promotion of tumourigenesis in the PTEN ko model and this should be epistatic to Arid1a loss. These experiments would be very time consuming in-vivo so I would suggest that the authors re-word their title and discussion to reflect the correlative links rather than cause and effect. Some in-vitro experiments suggested below should be informative.

Response: We agree and have modified the title and discussion (page 15, first paragraph) to describe the correlative studies and major findings rather than referring to “in vivo tumor suppressor of Arid1a”.

R4, Q2: 2) The mouse model is based on Arid1a and PTEN loss. It would be good to show the overlap and frequency of these events as an oncoprint of published NGS of human endometrial tumours.

Response: We now added analysis on the TCGA endometrial carcinoma which can be found in Figure 1B. The analysis indicates a significant co-occurrence between *ARID1A* and *PTEN* mutations.

R4, Q3: 3) In figures 1 and 2 the authors describe disseminated tumour nodules how do they know they are not multifocal separate initiation events? It would be good to stain these potential METS for PAX8 also. It would also be useful to provide Kaplan Meier analysis and quantification and location of disseminated disease in their model.

Response: We used a Pax8-Cre strategy which allows doxycycline-induced conditional knockout of *Arid1a* and/or *Pten* in organs/cells that express Pax8. In mouse, Pax8 is expressed specifically in thyroid, kidney, part of the central nervous system, inner ear, eye, and Wolffian and Müllerian ducts (PMID: 29113220). Our *iPAD* mouse displayed tumor dissemination in the intraperitoneal cavity. We have performed a detailed pathological assessment on the H&E slides from kidney of *iPAD* mice (six weeks after doxycycline treatment) that were excised at necropsy and did not observe any abnormality (Supplementary Figure 2B). These observations do not support the multifocal tumor initiation events because kidney as the only other organ that expresses Pax8 within peritoneal cavity uteri did not show any abnormality.

We have provided Kaplan Meier survival data analysis of *iPAD* mice with and without doxycycline (Figure 2D). We did not perform quantification of tumor dissemination because the widespread of the tumor in peritoneal cavity may lead to great variability and inaccuracy. Instead, we provide a representative image of the tumor at necrosy for this obvious finding (Figure 1C).

R4, Q4: 4) In Figure 3 the GSEA analysis and IPA analysis indicates that TGF β pathway is modulated in both the mouse model and the human cells lines. The authors should provide the GSEA analysis figure of the human data alongside that of the mouse (shown in Fig 3D). This may highlight that actually in the mouse the TGF β signaling pathway is enriched but in the human cell lines loss of Arid1a results in a down regulation of TGF β pathway signaling. The authors should comment on this.

Response: We now added GSEA analysis on TGFB1 target signature molecules (PLASARI_TGFB1_TARGETS and KEGG_TGF_BETA_SIGNALING) against the *ARID1A*^{WT}/*ARID1A*^{KO} human transcriptome shown in Supplementary Table 3 (lines 197-200 and Figure 3D). The plot showed the enrichment of TGF- β signaling in *ARID1A*^{WT} cells (FDR < 0.002), thus TGF- β signaling in *ARID1A*^{KO} cells were attenuated consistent with our biology data (Figure 4). In the mouse, we observed an enrichment of TGF- β signalling in *iPAD* mouse (FDR = 0.000). These opposite observations were probably due to the presence of immune cells in *iPAD* tumor (data not shown) which are *ARID1A* wild-type and responsive to TGF- β stimulation. Thus, enriching the TGF- β signalling in *iPAD* tumor.

R4, Q5: 5) A common event between the mouse and human RNAseq data is that TGFBR2 is downregulated. This should be investigated in more detail. Q-PCR validation in both sets of samples along with western blotting would be good. It would also be good to interrogate human RNAseq data from patient material to see if this also occurs during tumour progression and if this correlates with Arid1a and/or PTEN loss. To test if loss of TGFBR2 is responsible for a downregulation of exogenous TGF β response the authors should restore TGFBR2 levels to wild type levels in the human cell lines and measure the effect in PO4-Smad readouts.

Response: We now added (qPCR) validation data of TGFBR2 in human cells (Supplementary Figure 5A). Regarding interrogation of human RNAseq from patient material in relation to tumor progression, we feel that this seemingly large study deserves a separate publication in a pathology-oriented journal.

We have attempted to perform restoration experiment by transfecting TGFBR2 expression vector (Addgene: 11766). However, forced ectopic expression of TGFBR2 resulted in a significant amount of cell death. The average cell viability of the TGFBR2 over-expression experiments (n = 4) was 32.6%. Because of the high amount of cell death, we did not further pursue the checking of p-SMAD3 expression. We are not aware of any other method that can provide controlled restoration of protein expression to their wild-type levels.

Technical concerns

R4, Q6: 1) The authors should provide validation of their antibodies in IHC conditions for PO4-Smad3, Arid1a and PTEN by embedding cells +/- the antigens (knockout or loss for Arid1a, PTEN and +/- TGFBR inhibitor treated cells for PO4-Smad3) in FFPE conditions.

Response: The ARID1A antibody is highly specific as it stains only wild-type but not in mutated human tumors (PMID:29659191). The specificity of the other antibodies used in this study has also been previously reported (PTEN - PMID:21878536; p-SMAD3 - PMID:27189169).

R4, Q7: 2) The authors should provide more detail for how the murine tumours were classified into different grades with clear images of examples provided in the supplementary data.

Response: The grading system (FIGO) used here is the same as used in human endometrial cancer and this is performed by gynaecological pathologists. Further clarification is added in Histology and Immunohistochemistry section of the Methods (lines 762-765).

R4, Q8: 3) In Figure 3 the authors state that there is a reduction in intrinsic PO-Smad3 levels in the arid1a deficient mouse tissues. The images are unclear, blow up images to show nuclear PO4-Smad3 would be helpful and a histoscore quantification of reduction should be performed complete with statistical analysis. The baseline intrinsic signalling in the knockout human cells is not

statistically reduced and the statistical significance of the reduction in exogenous ligand stimulation should be analysed.

Response: We now added additional higher magnification inset in Supplementary Figure 4B to demonstrate weaker pSmad3 in *iAD* mice with induced *Arid1a*-deletion. Statistical analysis has been added in the revised manuscript.

R4, Q9: 4) In Supp Figure 4a the authors claim that restoration of ARID1A to ko cells restores TGFβ1 sensitivity. The data of the ARID1A ko cells +/- TGFβ1 stimulation, +/- Arid1A restoration should be shown in the invasion assays and assessed statistically.

Response: We now added the statistical analysis of the data in the Supplementary Figure 4C. Our control experiment (please see the figure below) using vector control and ARID1A expression plasmid in the absence of TGFβ1 demonstrated no significant difference in between the two conditions, as well as to the parental cells.

R4, Q10: 5) In Figure 4G the authors state that extracellular TGFβ1 induced a significant inhibition of forward migration in wt but not KO cells. This was not statistically assessed in the figure and should be.

Response: We now revised our statement as we observed a significant inhibition of forward migration in *ARID1A*^{WT} cells as compared to *ARID1A*^{KO} cells in the presence of extracellular TGF-β1 (lines 251-254).

Minor points/errors

R4, Q11: 1) Define UT in Figure 1B legend.

Response: We have added the definition.

R4, Q12: 2) Refer to the blue star in Figure 2b

Response: The star indicated tumor invasion into stroma (and is now provided in the legend).

R4, Q13: 3) The PCA analysis is not shown in Figure 5 and mislabelling/reference in the text to Figure 5 has occurred.

Response: This error has been corrected.

Reviewers' comments:

Reviewer #1 (Remarks to the Author):

The study is considerably improved in response to the four reviewers many comments and suggestions. Many points have been clarified, experiments improved, and wording carefully altered when appropriate to avoid potentially overstated claims. The reviewers have meticulously and thoroughly addressed the reviewer's comments in their rebuttal, making the needed changes or in a few cases thoughtfully arguing against the suggestions. Thus in my opinion this manuscript is now suitable for publication in Nature Communications.

Reviewer #2 (Remarks to the Author):

The authors have failed to address the majority of the points that I raised even when this required very little effort and no new experiments. See below my responses to the authors' rebuttal on points that they failed to address adequately.

R2, Q2: Figure 2: While experimental cohorts analyzed in Figure 2a are quite compelling, the authors should extend this data to other analyses, for example tumor weight and survival in iPAD, iPD, iAD, and controls. It is not sufficient to show survival +/- dox in the iPAD model alone. Response: We agree in principal and have indeed included iPAD +/- dox survival data (Figure 2D). We did not include survival data for iPD and iAD because both mice were alive at all time points tested (Figure 2A). With regards of tumor weight, in iPAD mouse tumor develop in the uteri and invade/spread into peritoneal cavity (Figure 1B). Thus, it would be highly challenging to accurately measure tumor weight in disseminated tumors. Instead, we provide a representative image of the tumor at necropsy.

Response to Rebuttal:

- 1) the authors could simply add the iPD and iAD + dox survival curves to 2D if they have them. These are important results that could be visualized rather than written in the text. It also seems likely that at least some of the iPD mice would succumb shortly after 24 weeks, information which would be good to include in a longer survival curve.
- 2) the authors show resected tumors in Figure 1C as a 'representative image'. What is pictured is something one could weigh.

R2, Q3: Figure 3: The RNA-seq analysis could be further developed here and some of the supplementary data represented in figure form. One thing that might help is to do RNA-seq from a normal control and derive differentially expressed genes between the control versus iPAD or iPD, as well as an iPAD versus iPD comparison. With the current visualization, the iPAD and iPD models look quite similar (e.g. by Euclidean distance in 3c and 3e), and yet have quite different kinetics of tumor development and metastasis. How many genes are differentially regulated between iPAD and iPD tumors? It would be helpful to include Venn Diagrams of DEGs between all comparisons, heat map clustering, and a list of the top 10 Hallmark enrichment, GO enrichment terms with p values for the mouse and human cell line data. As is, figure 3D is underwhelming...it is also not clear if this is a representative "TGF-b" gene set, or a one-off that happened to show enrichment.

Response: The Euclidian distance is a relative measurement between iPAD, iPD, and other human tumor (TCGA data). Therefore, iPAD and iPD models appear to be relatively similar to each another because we included a wide variety of tumor types. We identified 1,156 DEGs between iPAD and iPD tumors (please see lines 163-166 and Supplementary Table 1). Moreover, we performed "Ingenuity IPA top upstream network regulators analysis" on our mouse and human DEGs to identify TGFB1 were shared between the mouse and human networks (please see lines 166-169, 193-197, and Supplementary Table 2 and 4) and have included the p-value.

Response to Rebuttal:

I appreciate that the Euclidian distance will be similar; however, this analysis also clearly reflects that this measurement does not have sufficient resolution to capture differences associated with the pathogenic nature of iPAD versus iPD. The mouse and human have inverse enrichments for the TGF- β pathway despite the fact that this came up in the IPA analysis for both. That is why it would be useful to 'include Venn Diagrams of DEGs between all comparisons, heat map clustering, and a list of the top 10 Hallmark enrichment, GO enrichment terms with p values for the mouse and human cell line data.' This was not done despite the fact that this data is in hand.

R2, Q4: Also, now that RNA-seq data is available for primary endometrial cancers through the TCGA, there is no need to rely on CRISPR-derived cell lines. The authors should use publically available data and perform analyses against their GEM model to see how representative the model is. It would be necessary to provide a complete work-up here of the mouse to human primary tumor comparison: GSEA, GO pathway terms, leading edge genes identified, etc.

Response: To determine the role of ARID1A mutations in endometrial cancer carcinogenesis, we used ARID1A CRISPR knockout on primary human endometrial epithelial cells to mimic ARID1A mutations which lead to loss of ARID1A protein. We hope that this provides a "clean" model (only ARID1A is deleted) which complements to the TCGA tumors which harbor a variety of genetic alterations in addition to ARID1A mutation. To answer the 2nd question, we have used publically available data and performed Euclidian distance analysis on our mouse RNA-seq data and demonstrated that our GEM model is most similar to UCEC dataset with respect to 7 TCGA carcinomas used in the analysis (please see Figure 3B and 3C).

Response to Rebuttal:

This was not done or even attempted for key genes in this analysis. Using the cBio portal, it would be very easy to monitor the expression of key genes in ARID1A altered versus unaltered for the genes represented in Figure 5E, or attempt the more comprehensive analysis as suggested.

R2, Q8: Figure 5. The ChIP-data and ATAC-seq data look quite well done. However, the analysis of these data could be expanded significantly. Previous reports have shown that ARID1A can both positively regulate enhancers to promote gene expression (Mathur et al Nat Genetics 2017, Kelso Nat Comm 2017) and also repress genes through recruitment of HDACs primarily at promoters (Kim Cell Reports 2016). To define the mechanism of ARID1A action, the authors should analyze where in the genome ARID1A is most commonly bound (promoters, enhancers, etc), as well as TF motifs underlying ARID1A binding sites. They should perform a comparative analysis of ARID1A loss and ATAC-seq loss (how many sites, which TF motifs, etc). Also, are there any ATAC-seq gains? How do ATAC-seq gains compare with ARID1A binding in the WT? How do ATAC loss and gains relate to changes in gene expression?

Response: As suggested, we have added more analyses on ARID1A binding location and TF. Comparing ATAC-seq signals before and after ARID1A knockout, we identified 15,192 (38.6%) of ATAC-seq peak losses, of which 8,979 (~60%) peaks were associated with ARID1A-specific binding events (please see figure below). motifs (Supplementary Figure 5C-D; lines 273-275; lines 277-280). We also found that 496 ATAC-seq signals were gained in response to ARID1A loss, of which 33 peaks were associated with ARID1A-specific peaks (please see figure below). Moreover, ATAC loss was more significantly associated with differential genes after ARID1A knockout than the ATAC gain.

Response to Rebuttal:

This is very relevant to the study and should be included in a Figure or Supplemental figure. Ideally, an overlap of ARID1A-dependent ATAC peaks, ARID1A-dependent BRG1 peaks, ARID1A ChIP binding sites, and ARID1A-dependent genes would be good with location information (enhancer versus promoter). The authors have already done the analysis to say that ARID1A binding is enriched at ARID1A-dependent ATAC peaks, ARID1A-dependent BRG1 peaks, and

ARID1A-dependent genes, and that loss in ATAC is associated with changes in gene expression. I'm not sure why the authors chose to leave out this information and not add a Figure panel or Supplemental panel. From their response to the reviewer, it appears that they favor the hypothesis that ARID1A is regulating gene expression by promoting accessibility at distal sites. Previous reports have suggested that ARID1A is acting to suppress genes related to inflammation and EMT by binding directly to the promoters of these genes and suppressing remodeling (Kim Cell Reports 2016, Chandler Nat Comm 2015, Wilson Nat Comm 2019). Indeed, Wilson et al. 2019 claim that in their endometrial model of ARID1A KO/PIK3CA H1047R that ARID1A is directly binding the promoters of EMT genes and observe an increase in ATAC-seq at the promoters of these genes upon ARID1A deletion. It is thus very timely and relevant if the authors are in fact not observing this as the primary means of ARID1A transcriptional regulation. Figures clarifying this point are necessary.

Reviewer #3 (Remarks to the Author):

Reviewer-3 response to rebuttals

The authors have responded satisfactorily to the majority of this Reviewer's comments. A few remaining suggestions are as follows:

R3_Q3. Suggest changing the revised sentence in the Summary to "we analyzed ARID1AWT and ARID1AKO human endometrial epithelial cells."

R3_Q5. The revised sentence on Lines 54-56 ("uterine endometrioid carcinoma affects 63,000 women annually and claims the lives of more than 11,000 women each year in the United States") is inaccurate. The 63,000 new cases and 11,000 deaths refers to all uterine cancers (ie. endometrial carcinomas, of 12 different histological subtypes, and uterine sarcomas). Uterine endometrioid endometrial cancer accounts for just a fraction of these new cases and deaths. I suggest modifying this sentence to read "Currently, uterine cancer affects 63,000 women annually and claims the lives of more than 11,000 women each year in the United States 13". I also suggest making the following revisions to line 61, so that it reads: "The majority of uterine cancers are endometrial cancers. Endometrial cancer can be classified into different subtypes with endometrioid and serous adenocarcinoma being the most common (X% and X% of newly diagnosed cases respectively)." The authors will need to provide the value (X) for the percentage in the last sentence.

R3_Q6 and Q7. The first part of the revised text ("Previous studies reported a much frequent occurrence of ARID1A and PTEN (or PIK3CA") mutation in endometrioid subtype of endometrial cancer than in the serous subtype) is okay and addresses Q6 of this reviewer. The second part of this sentence ("suggesting cooperation between these two gene pathways") is inaccurate because functional co-operativity cannot be inferred based on frequency alone. I suggest modifying this text to read "Previous studies reported a higher frequency of ARID1A and PTEN (or PIK3CA) mutations in the endometrioid subtype of endometrial cancer than in the serous subtype. Moreover, ARID1A and PTEN mutations frequently co-occur in endometrioid endometrial tumors 7, 11."

Reviewer #4 (Remarks to the Author):

I thank the authors for carefully addressing my concerns in their revised manuscript which has resulted in an improved paper. I still have a few suggestions that I would like the authors to

address

1) The authors have acknowledged the opposite regulation of TGFb signaling pathways in the mouse tumors and human cell lines in their response to my comments along with a possible explanation but they have not included this discussion in their manuscript. They should highlight this in the paper when they refer to figure 3 as this is an important point. In the mouse tumors TGFb1 ligand is elevated but this is not the case in the cell lines so an enrichment of TGFb signaling in tumors could be due to stromal TGFb signaling. Importantly in both data sets TGFBR2 is downregulated indicating a tumor cell intrinsic reduction in TGFb response. This should be discussed in much more detail

2) In response to my suggested TGFBR2 restoration experiments the authors indicate in their response that they have attempted these experiments but this induces cell death. This is a potentially important observation that may go some way to explaining their phenotypes. The authors current view expressed in their manuscript is that loss of TGFb responsiveness following Arid1a deletion is responsible for an increased migratory and invasive phenotype. The figure supplied to reviewers showing the effects of Arid1a restoration in KO cells does indicate restoration of an inhibitory TGFb response in this assay but does not indicate that the differences in invasion between ko cells (which have very poor migration and invasion) and wt cells which have good invasion and migration is explained by this. The authors should treat the Arid1a wt and KO cells -/+ Arid1a reexpression with SB431542 to inhibit endogenous TGFb signaling and determine the effects on both invasion in the Xcelligence assays and on cell migration in the wound heal assays. If inhibiting TGFb signaling has no effect then the profound differences in basal migration and invasion observed following Arid1a deletion are not due changes in TGFb response. The effect of restoring TGFBR2 on cell survival is intriguing so the authors should also perform the cell proliferation and viability experiments -/+ TGFb and -/+ SB431542 shown in Figure 4D and then also perform these experiments -/+ TGFBR2 restoration in the Arid1a Ko cells. It could be in the absence of Arid1a endometrial cells become susceptible to TGFb mediated cell death and hence the requirement to downregulate the receptor.

Technical points

1)The blow up images of PO4-Smad3 IHC Show in Supp Figure 4 B should be used in Fig 4A

2) Khalique et al validation of Arid1a IHC paper should be cited in the text similarly Lotan et al validation of PTEN IHC should also be cited in the text. Chen et al referred to in authors response to reviewers does not validate PO4-Smad3 IHC a better example is PMID: 27558455.

Typos

Page 2 Through a systems biology approach

Page 12 TGFBR2 by in (delete by)

Page 14 were presented in Fig 5D should be Fig 5E

Page 15 including the TGFb receptor should be TGFBR2

Reviewer #1 (Remarks to the Author): Expertise in endometrial cancer models

The study is considerably improved in response to the four reviewers many comments and suggestions. ... meticulously and thoroughly addressed the reviewer's comments in their rebuttal, making the needed changes or in a few cases thoughtfully arguing against the suggestions. Thus in my opinion this manuscript is now suitable for publication in Nature Communications.

Reviewer #2 (Remarks to the Author): Expertise in ARID1A

In response to R2, Q2 (1st revision) on tumor weight and survival in iPAD, iPD, iAD, and controls mice:

R2, Q1 (this revision): the authors could simply add the iPD and iAD + dox survival curves to 2D if they have them. These are important results that could be visualized rather than written in the text. It also seems likely that at least some of the iPD mice would succumb shortly after 24 weeks, information which would be good to include in a longer survival curve.

Response: As per the editorial suggestion, we did not perform in vivo experiment. We will consider a similar study in the future.

R2, Q2 (this revision): the authors show resected tumors in Figure 1C as a 'representative image'. What is pictured is something one could weigh.

Response: As per the editorial suggestion, we did not perform in vivo experiment.

In response to R2, Q2 (1st revision) on Venn Diagrams of DEGs between all comparisons, heat map clustering, and a list of the top 10 Hallmark enrichment, GO enrichment terms:

R2, Q3 (this revision): I appreciate that the Euclidian distance will be similar; however, this analysis also clearly reflects that this measurement does not have sufficient resolution to capture differences associated with the pathogenic nature of iPAD versus iPD. The mouse and human have inverse enrichments for the TGF- β pathway despite the fact that this came up in the IPA analysis for both.

Response: We performed Euclidian distance analysis to assess molecular relevance of our mouse tumors to corresponding human tumors (please see line 165-169, page 8). We do not intend to use Euclidian distance analysis to "capture differences associated with the pathogenic nature of iPAD versus iPD", as inferred by the reviewer.

We have now included possible explanation on the inverse enrichment of TGF- β signalling when we refer to Figure 3D in line 197-202, page 9-10.

R2, Q4 (this revision): That is why it would be useful to 'include Venn Diagrams of DEGs between all comparisons, heat map clustering, and a list of the top 10 Hallmark enrichment, GO enrichment terms with p values for the mouse and human cell line data.' This was not done despite the fact that this data is in hand.

Response: We have now included Venn diagrams of DEGs in mouse and human transcriptomes (Figure 3E), showing the overlap between groups. GO enrichment terms commonly identified in ARID1A-associated transcriptomes of mouse uterine tumors and human endometrial epithelial cells is now added in Supplementary Table 5. In the manuscript, we performed “Ingenuity IPA top upstream network regulators analysis” on our mouse and human DEGs to identify TGFB1 were shared between the mouse and human networks (please see lines 162-164, 191-195, and Supplementary Table 2 and 4) and have included the p-value.

In the revised submission, we did not include the Hallmark enrichment analyses as suggested because the analyses were not informative and did not produce significant/consistent enrichment (almost all FDR q-values are > 0.05, please see Table R1 and R2 below).

Table R1. Hallmark enrichment analysis on the DEGs identified in two pairs of isogenic *ARID1A^{WT}* and *ARID1A^{KO}* human endometrial epithelial cells.

NAME	SIZE	ES	NES	FDR q-val
Enrichment in ARID1A^{WT}				
HALLMARK_TNFA_SIGNALING_VIA_NFKB	22	0.445	1.770	0.095
HALLMARK_EPITHELIAL_MESENCHYMAL_TRANSITION	30	0.386	1.716	0.070
HALLMARK_IL2_STAT5_SIGNALING	20	0.381	1.495	0.177
HALLMARK_KRAS_SIGNALING_UP	31	0.281	1.273	0.365
HALLMARK_ESTROGEN_RESPONSE_LATE	23	0.231	0.967	0.808
HALLMARK_INTERFERON_GAMMA_RESPONSE	16	0.261	0.927	0.741
HALLMARK_UV_RESPONSE_DN	22	0.228	0.900	0.675
HALLMARK_ESTROGEN_RESPONSE_EARLY	30	0.147	0.651	0.886
Enrichment in ARID1A^{KO}				
HALLMARK_HYPOXIA	23	-0.315	-1.459	0.386
HALLMARK_MYOGENESIS	20	-0.319	-1.395	0.259
HALLMARK_MTORC1_SIGNALING	15	-0.205	-0.811	1.000
HALLMARK_APICAL_JUNCTION	24	-0.164	-0.758	0.978
HALLMARK_P53_PATHWAY	19	-0.166	-0.710	0.843

Table R2. Hallmark enrichment analysis on the DEGs identified between *iPD* and *iPAD* uterine tumors.

NAME	SIZE	ES	NES	FDR q-val
Enrichment in iPD				
HALLMARK_EPITHELIAL_MESENCHYMAL_TRANSITION	46	0.343	1.489	0.368
HALLMARK_KRAS_SIGNALING_DN	18	0.443	1.469	0.197
HALLMARK_MYOGENESIS	22	0.346	1.234	0.455
HALLMARK_ADIPOGENESIS	25	0.320	1.169	0.455
HALLMARK_COAGULATION	20	0.294	1.019	0.681
HALLMARK_APICAL_JUNCTION	30	0.259	0.980	0.647
HALLMARK_APOPTOSIS	19	0.274	0.941	0.623
HALLMARK_ESTROGEN_RESPONSE_LATE	41	0.206	0.881	0.648
Enrichment in iPAD				
HALLMARK_MTORC1_SIGNALING	18	-0.637	-1.828	0.021
HALLMARK_INFLAMMATORY_RESPONSE	33	-0.479	-1.580	0.161
HALLMARK_FATTY_ACID_METABOLISM	16	-0.331	-0.914	0.671
HALLMARK_HYPOXIA	32	-0.280	-0.918	0.711
HALLMARK_XENOBIOTIC_METABOLISM	34	-0.281	-0.928	0.749
HALLMARK_ESTROGEN_RESPONSE_EARLY	35	-0.293	-0.982	0.770
HALLMARK_COMPLEMENT	29	-0.323	-1.009	0.782
HALLMARK_INTERFERON_GAMMA_RESPONSE	21	-0.398	-1.190	0.810
HALLMARK_ALLOGRAFT_REJECTION	22	-0.311	-0.928	0.811
HALLMARK_TNFA_SIGNALING_VIA_NFKB	35	-0.344	-1.143	0.821

In response to R2, Q4 (1st revision) on RNA-seq data is available for primary endometrial cancers through the TCGA, there is no need to rely on CRISPR-derived cell lines:

R2, Q5 (this revision): This was not done or even attempted for key genes in this analysis. Using the cBio portal, it would be very easy to monitor the expression of key genes in ARID1A altered versus unaltered for the genes represented in Figure 5E, or attempt the more comprehensive analysis as suggested.

Response: We have now done this analysis and included z-score correlation between *ARID1A* mRNA expression and 'genes in TGF- β signaling pathway that are directly regulated by ARID1A', please see line 323-327, page 14 and Supplementary Figure 5G.

We have also attempted to perform the analysis using categorical method of 'ARID1A altered versus unaltered' (as kindly suggested by Reviewer 2) but found that the available data on TCGA UCEC PanCancer Atlas did not show correlation between *ARID1A* mutation and *ARID1A* mRNA expression ($p = 0.1653$, please see figure below), most likely due to the fact that TCGA tumors, unlike cell lines, contained non-tumor cells such as stromal cells, immune cells and endothelial cells, etc. which express ARID1A and were included in the expression analysis.

In response to R2, Q8 (1st revision) on a comparative analysis of ARID1A loss and ATAC-seq loss (how many sites, which TF motifs, etc):

R2, Q6 (this revision): This is very relevant to the study and should be included in a Figure or Supplemental figure.

Response: We have now included comparative analysis of ATAC-seq signals in ARID1A loss as Supplementary Figure 5F.

R2, Q7 (this revision): Ideally, an overlap of ARID1A-dependent ATAC peaks, ARID1A-dependent BRG1 peaks, ARID1A ChIP binding sites, and ARID1A-dependent genes would be good with location information (enhancer versus promoter). The authors have already done the analysis to say that ARID1A binding is enriched at ARID1A-dependent ATAC peaks, ARID1A-dependent BRG1 peaks, and ARID1A-dependent genes, and that loss in ATAC is associated with

changes in gene expression. I'm not sure why the authors chose to leave out this information and not add a Figure panel or Supplemental panel. From their response to the reviewer, it appears that they favor the hypothesis that ARID1A is regulating gene expression by promoting accessibility at distal sites. Previous reports have suggested that ARID1A is acting to suppress genes related to inflammation and EMT by binding directly to the promoters of these genes and suppressing remodeling (Kim Cell Reports 2016, Chandler Nat Comm 2015, Wilson Nat Comm 2019). Indeed, Wilson et al. 2019 claim that in their endometrial model of ARID1A KO/PIK3CA H1047R that ARID1A is directly binding the promoters of EMT genes and observe an increase in ATAC-seq at the promoters of these genes upon ARID1A deletion. It is thus very timely and relevant if the authors are in fact not observing this as the primary means of ARID1A transcriptional regulation. Figures clarifying this point are necessary.

Response: As noted by the reviewer, there are conflicting literatures with some suggesting that 'ARID1A regulate genes by binding directly to promoter' (Kim Cell Reports 2016, Chandler Nat Comm 2015, Wilson Nat Comm 2019), while others suggested that 'ARID1A has role in maintaining chromatin accessibility at enhancers' (Kelso eLife 2017, Mathur Nat Genetics 2017, Sun Cancer Cell 2017, Sen Clin Epigenetics 2019).

As it stands, our data suggested that ARID1A can bind to both promoter and enhancer regions (Supplementary Fig. 5C). It is possible that ARID1A can work on promoter and/or enhancer depending on tissue origin, cell types, and the nature of microenvironment. Moreover, our ATAC-/ChIP-seq data is a single time-point experiment, and we believe that a multiple time-point kinetic type experiment would be more appropriate to provide an answer to this question. Although we agree that this is a good idea, we also recognize that this would require a more careful, extensive and meticulous analyses, to better address this issue. Thus, we feel that this can be beyond the scope of our current manuscript aiming at providing timely information about how "Inactivation of *Arid1a* in endometrium promotes tumorigenesis through transcription regulation of TGF- β Signaling Pathway". We plan to perform such experiments with a comprehensive design and deeper analysis in an independent study in the near future.

Reviewer #3 (Remarks to the Author): expertise in endometrial cancer

The authors have responded satisfactorily to the majority of this Reviewer's comments. A few remaining suggestions are as follows:

R3, Q1 (this revision): R3_Q3. Suggest changing the revised sentence in the Summary to "we analyzed ARID1AWT and ARID1AKO human endometrial epithelial cells."

Response: We agree and have revised the sentence accordingly, please see line 31-32, page 2.

R3, Q2 (this revision): R3_Q5. The revised sentence on Lines 54-56 ("uterine endometrioid carcinoma affects 63,000 women annually and claims the lives of more than 11,000 women each year in the United States") is inaccurate. The 63,000 new cases and 11,000 deaths refers to all uterine cancers (ie. endometrial carcinomas, of 12 different histological subtypes, and uterine sarcomas). Uterine endometrioid endometrial cancer accounts for just a fraction of these new cases and deaths. I suggest modifying this sentence to read "Currently, uterine cancer affects 63,000 women annually and claims the lives of more than 11,000 women each year in the United States 13". I also suggest making the following revisions to line 61, so that it reads: "The majority

of uterine cancers are endometrial cancers. Endometrial cancer can be classified into different subtypes with endometrioid and serous adenocarcinoma being the most common (X% and X% of newly diagnosed cases respectively).” The authors will need to provide the value (X) for the percentage in the last sentence.

Response: We agree and have revised the two sentences and updated the numbers according to the most recent reference, please see line 53-54 and 59-61, page 3.

R3, Q3 (this revision): R3_Q6 and Q7. The first part of the revised text (“Previous studies reported a much frequent occurrence of ARID1A and PTEN (or PIK3CA”) mutation in endometrioid subtype of endometrial cancer than in the serous subtype) is okay and addresses Q6 of this reviewer. The second part of this sentence (“suggesting cooperation between these two gene pathways”) is inaccurate because functional co-operativity cannot be inferred based on frequency alone. I suggest modifying this text to read “Previous studies reported a higher frequency of ARID1A and PTEN (or PIK3CA) mutations in the endometrioid subtype of endometrial cancer than in the serous subtype. Moreover, ARID1A and PTEN mutations frequently co-occur in endometrioid endometrial tumors 7, 11.”

Response: We agree and have revised the sentence accordingly, please see line 61-63, page 3.

Reviewer #4 (Remarks to the Author): expertise in TGFbeta signalling

I thank the authors for carefully addressing my concerns in their revised manuscript which has resulted in an improved paper. I still have a few suggestions that I would like the authors to address:

R4, Q1 (this revision): The authors have acknowledged the opposite regulation of TGFb signaling pathways in the mouse tumors and human cell lines in their response to my comments along with a possible explanation but they have not included this discussion in their manuscript. They should highlight this in the paper when they refer to figure 3 as this is an important point. In the mouse tumors TGFB1 ligand is elevated but this is not the case in the cell lines so an enrichment of TGFb signaling in tumors could be due to stromal TGFb signaling. Importantly in both data sets TGFBR2 is downregulated indicating a tumor cell intrinsic reduction in TGFb response. This should be discussed in much more detail

Response: We agree and have now included possible explanation on the opposite enrichment of TGF-β signalling when we refer to Figure 3D in line 197-202, page 9-10.

R4, Q2 (this revision): In response to my suggested TGFBR2 restoration experiments the authors indicate in their response that they have attempted these experiments but this induces cell death. This is a potentially important observation that may go some way to explaining their phenotypes. The authors current view expressed in their manuscript is that loss of TGFb responsiveness following Arid1a deletion is responsible for an increased migratory and invasive phenotype. The figure supplied to reviewers showing the effects of Arid1a restoration in KO cells does indicate restoration of an inhibitory TGFb response in this assay but does not indicate that the differences in invasion between ko cells (which have very poor migration and invasion) and wt cells which have good invasion and migration is explained by this. The authors should treat the Arid1a wt and KO cells +/- Arid1a reexpression with SB431542 to inhibit endogenous TGFb

signaling and determine the effects on both invasion in the Xcelligence assays and on cell migration in the wound heal assays. If inhibiting TGF β signaling has no effect then the profound differences in basal migration and invasion observed following Arid1a deletion are not due changes in TGF β response.

Response: We appreciate reviewer's thoughtful consideration and suggestion. We have now treated *ARID1A*^{WT} and *ARID1A*^{KO} cells with SB431542 (1 μ M and 10 μ M). We found that *ARID1A*^{WT} cells had enhanced invasion through the extracellular matrix (xCELLigence, left panel) after inhibitor treatment, while in *ARID1A*^{KO} cells (right panel), a significant effect on cell invasion can only be observed at 10 μ M but not at 1 μ M.

Our transcriptome data showed that deletion of ARID1A led to downregulation of TGFBR2 and consequently suppressed TGF- β signalling. Thus, it is expected that *ARID1A*^{WT} will have the basal sensitivity towards blockage of TGF- β signalling, while *ARID1A*^{KO}, which has already reduced TGF- β signalling, will not be as responsive to TGF- β inhibitors. Altogether, our data suggest that ARID1A deletion has some effect in TGF- β dependent cellular invasion. We have incorporated our new results on page 11-12, lines 248-251.

Cell invasion capacity measured in *ARID1A*^{WT} (left panel) and *ARID1A*^{KO} cells (right panel) treated with 1 μ M and 10 μ M of TGF- β signaling inhibitor, SB431542. Cell invasion assays were performed using the xCELLigence RTCA real-time monitoring system. Data are expressed as mean \pm SEM (n = 3). *p<0.05; n.s., not significant; as determined by one-way ANOVA with Bonferroni's multiple comparison post-test by comparing two groups over time.

R4, Q3 (this revision): The effect of restoring TGFBR2 on cell survival is intriguing so the authors should also perform the cell proliferation and viability experiments +/- TGF β and +/- SB431542 shown in Figure 4D and then also perform these experiments +/- TGFBR2 restoration in the Arid1a Ko cells. It could be in the absence of Arid1a endometrial cells become susceptible to TGF β mediated cell death and hence the requirement to downregulate the receptor.

Response: We thank the reviewer for the suggestion. We have now performed cell proliferation/viability experiments on *ARID1A*^{WT} (left panel) and *ARID1A*^{KO} (right panel) cells with and without TGF- β ligand and SB431542 inhibitor. In our experimental conditions, we did not observe any significant difference (as determined using one-way ANOVA with Bonferroni's multiple comparison post-test by comparing two groups over time) between the parental untreated cells and all of treatment groups.

Our data suggest that the loss of ARID1A cause endometrial epithelial cells to become more resistance (instead of the reviewer suggestion of 'susceptible') to TGF- β mediated cell death, due to the reduced *TGFBR2* expression, which is an ARID1A target genes. This is in agreement with our results on restoring TGFBR2 in *ARID1A^{WT}* and *ARID1A^{KO}* cells, which led to enhanced TGF- β signalling and its physiological consequence of cell death.

Measurement of cellular proliferation for *ARID1A^{WT}* (left panel) and *ARID1A^{KO}* (right panel) cells over 48 hours. Data are expressed as mean \pm SEM (n = 3). Statistical analysis was performed using one-way ANOVA with Bonferroni's multiple comparison post-test by comparing two groups over time.

R4, Q4 (this revision): The blow up images of PO4-Smad3 IHC Show in Supp Figure 4 B should be used in Fig 4A

Response: We agree and have now swapped the Supplementary Figure 4B with Figure 4A.

R4, Q5 (this revision): Khalique et al validation of Arid1a IHC paper should be cited in the text similarly Lotan et al validation of PTEN IHC should also be cited in the text. Chen et al referred to in authors response to reviewers does not validate PO4-Smad3 IHC a better example is PMID: 27558455.

Response: We agree and have now cited these recommended references of Khalique PMID:29659191, Lotan PMID:21878536, and Cammareri PMID:27558455, please see line 794-799, page 33.

R4, Q6 (this revision): Typos

Page 2 Through a systems biology approach

Page 12 TGFBR2 by in (delete by)

Page 14 were presented in Fig 5D should be Fig 5E

Page 15 including the TGF β receptor should be TGFBR2

Response: We agree and have corrected these typographical errors accordingly.

REVIEWERS' COMMENTS:

Reviewer #2 (Remarks to the Author):

The authors have now addressed all of my comments.

Reviewer #4 (Remarks to the Author):

I thank the reviewers for further considering my points which has improved the manuscript. I still however have some questions regarding a central conclusion that loss of Arid1a results in a "loss of TGFb tumor suppressor function and inactivation of Arid1a/TGFb axis promotes migration and invasion of PTEN deleted endometrial tumor cells"

The authors provide convincing evidence that loss of Arid1a results in a downregulation of TGFBR2 and a reduction in endogenous and exogenous PO4-Smad3 activity in vitro and in-vivo. The authors also show that exogenous TGFb can inhibit migration of wt cells in a wound heal scratch assay but not in ko cells (Fig 4E). They have not performed this assay as shown in this figure with and without SB431542 to determine if the enhanced migration rate seen in KO cells can be mimicked by SB431542 treatment of wt cells as previously asked. If it cannot then the enhanced migration rate may be due to other factors.

In terms of invasion then the xcelligence figure shown in Figure 4F shows a large difference in invasion rates between ko and wt cells. In addressing the effects of endogenous TGFb signaling now provided in supp Fig 4d then this profound difference is very much diminished. This questions the reproducibility of the invasion differences. I understand that invasion rates may differ from batch to batch of matrigel but differences between wt and KO cells should remain reasonably consistent. The authors should plot the SB431542 data with WT and KO cells on the same graph and then determine if WT cells are statistically significantly less invasive than KO cells and if SB431542 treatment then enables WT cells to invade at the same rate as KO cells. This should be performed several times to ensure reproducibility.

In interpreting the XFMI data the authors say TGFb inhibits this in WT cells and not in KO cells. The authors should statistically analyse this +/- TGFb in wt cells and +/- TGFb in KO cells, the analysis presented indicates that in the presence of TGFb the XFMI is greater in KO cells compared to WT cells.

On Line 347 the authors state that "we demonstrated that TGFb contributed to the in-vivo tumor suppressor function of Arid1a". They have not shown this.

Line 191 "839" should be "893"

Reviewer #4: expertise in TGFbeta signalling

R4, Q1 (this revision): The authors provide convincing evidence that loss of Arid1a results in a downregulation of TGFBR2 and a reduction in endogenous and exogenous PO4-Smad3 activity in vitro and in-vivo. The authors also show that exogenous TGFb can inhibit migration of wt cells in a wound heal scratch assay but not in ko cells (Fig 4E). They have not performed this assay as shown in this figure with and without SB431542 to determine if the enhanced migration rate seen in KO cells can be mimicked by SB431542 treatment of wt cells as previously asked. If it cannot then the enhanced migration rate may be due to other factors.

Response: With the limited access to our laboratory during the COVID-19 pandemics, we have tried our best to perform the wound heal scratch experiment with and without the addition of SB431542 inhibitor. In the presence of TGFβ1 ligand, the addition of SB431542 show a trend of increased wound closure rate in *ARID1A^{WT}* cells, although not statistically significant (obtained from a single experiment, $p = 0.097$, slope comparison - linear regression test). As expected, the closure rate of *ARID1A^{KO}* cells was not affected by SB431542 ($p = 0.983$, slope comparison - linear regression test). We have described this preliminary finding in the text (please see line 254-258) and as Supplementary Fig. 4D. The above data are based on the wound heal scratch assay. Importantly, using a more sophisticated xCelligence assay, we observed that SB431542 was able to enhance *ARID1A^{WT}* cell invasion to a statistically similar level as *ARID1A^{KO}* (please response to R4, Q2 below).

R4, Q2 (this revision): In terms of invasion then the xcelligence figure shown in Figure 4F shows a large difference in invasion rates between ko and wt cells. In addressing the effects of endogenous TGFb signaling now provided in supp Fig 4d then this profound difference is very much diminished. This questions the reproducibility of the invasion differences. I understand that invasion rates may differ from batch to batch of matrigel but differences between wt and KO cells should remain reasonably consistent. The authors should plot the SB431542 data with WT and KO cells on the same graph and then determine if WT cells are statistically significantly less invasive than KO cells and if SB431542 treatment then enables WT cells to invade at the same rate as KO cells. This should be performed several times to ensure reproducibility.

Response: We have merged and shown the combined plot in Supplementary Fig. 4E. In the presence of exogenous TGFβ1, *ARID1A^{KO}* cells were more invasive than *ARID1A^{WT}* cells (magenta vs dark green lines, $p < 0.001$, one-way ANOVA with Bonferroni's multiple comparison post-test by comparing two groups over time), consistent with the results shown in Figure 4F. Indeed, we have performed this experiment more than 3 times with a similar result. We have also included additional text (please see line 261-263) describing that SB431542 treatment increased the *ARID1A^{WT}* cell invasion rates to a comparable level of *ARID1A^{KO}*.

R4, Q3 (this revision): In interpreting the XFMI data the authors say TGFb inhibits this in WT cells and not in KO cells. The authors should statistically analyse this +/- TGFb in wt cells and +/- TGFb in KO cells, the analysis presented indicates that in the presence of TGFb the XFMI is greater in KO cells compared to WT cells.

Response: The single-cell tracking experiment was designed to ask what we believed the most critical question: whether the presence of extracellular TGFβ1 resulted in a significant xFMI difference between in *ARID1A^{WT}* and *ARID1A^{KO}* cells. We then later performed a control experiment to compare *ARID1A^{WT}* and *ARID1A^{KO}* cells in the absence of TGFβ1. Thus, it would be challenging to directly compare *ARID1A^{WT}* cells +/- TGFβ1 because this set of experiment was not done at the same time as the previous experiment. Therefore, as suggested by the reviewer, we have now revised our statement to

better reflect our presented data and the question we intended to ask (please see line 272-273):
“However, the presence of extracellular TGF- β 1 resulted in a significant xFMI difference between *ARID1A*^{WT} and *ARID1A*^{KO} cells (Fig. 4H).”

R4, Q4 (this revision): On Line 347 the authors state that "we demonstrated that TGFb contributed to the in-vivo tumor supressor function of Arid1a". They have not shown this.

Response: We have now deleted the sentence on line 347, as it was misplaced. The same sentence was rephrased on line 403-405.

R4, Q5 (this revision): Line 191 "839" should be "893"

Response: We have corrected this error.